# Highway Graph to Accelerate Reinforcement Learning

**Zidu Yin**                                                        *zidu.yin@ynnu.edu.cn*
*School of Information Science and Technology*
*Yunnan Normal University*

**Zhen Zhang**                                                        *zhen@zzhang.org*
*School of Computer and Mathematical Sciences*
*Adelaide University*

**Dong Gong**                                                        *edgong01@gmail.com*
*School of Computer Science and Engineering*
*The University of New South Wales*

**Stefano V. Albrecht**                                              *s.albrecht@ed.ac.uk*
*School of Informatics*
*University of Edinburgh*

**Javen Q. Shi**                                                    *javen.shi@adelaide.edu.au*
*School of Computer and Mathematical Sciences*
*Adelaide University*

**Reviewed on OpenReview:** *https://openreview.net/forum?id=3mJZfL77WM*

## Abstract

Reinforcement Learning (RL) algorithms often struggle with low training efficiency. A common approach to address this challenge is integrating model-based planning algorithms, such as Monte Carlo Tree Search (MCTS) or Value Iteration (VI), into the environmental model. However, VI faces a significant limitation: it requires iterating over a large tensor with dimensions $|\mathcal{S}| \times |\mathcal{A}| \times |\mathcal{S}|$, where $\mathcal{S}$ and $\mathcal{A}$ represent the state and action spaces, respectively. This process updates the value of the preceding state $s_{t-1}$ based on the succeeding state $s_t$ through value propagation, resulting in computationally intensive operations. To enhance the training efficiency of RL algorithms, we propose improving the efficiency of the value learning process. In deterministic environments with discrete state and action spaces, we observe that on the sampled empirical state-transition graph, a non-branching sequence of transitions—termed a *highway*—can take the agent directly from $s_0$ to $s_T$ without deviation through intermediate states. On these non-branching highways, the value-updating process can be streamlined into a single-step operation, eliminating the need for iterative, step-by-step updates. Building on this observation, we introduce a novel graph structure called the *highway graph* to model state transitions. The highway graph compresses the transition model into a compact representation, where edges can encapsulate multiple state transitions, enabling value propagation across multiple time steps in a single iteration. By integrating the highway graph into RL (as a model-based off-policy RL method), the training process is significantly accelerated, particularly in the early stages of training. Experiments across four categories of environments demonstrate that our method learns significantly faster than established and state-of-the-art model-free and model-based RL algorithms (often by a factor of 10 to 150) while maintaining equal or superior expected returns. Furthermore, a deep neural network-based agent trained using the highway graph exhibits improved generalization capabilities and reduced storage costs. The implementation of our highway graph RL method is publicly available at https://github.com/coodest/highwayRL.

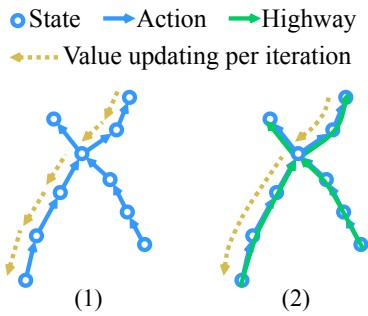

Figure 1: Comparison of the state-transition graph and its corresponding highway graph. An RL agent on the state-transition graph learns/propagates values state-by-state per learning step, and the highway graph propagates values for a stack of states in the highway per learning step. The highway graph is much smaller than the state-transition graph which leads to more efficient value learning. Take the Atari game Star Gunner as an example, the 234,521 states in the sampled empirical state-transition graph are represented by highway graph with only 1,084 (0.5% of its original size) states connected by highways.

# 1 Introduction

Reinforcement learning (RL) agent training is often time-intensive, primarily due to the low efficiency of the value-updating process. From the perspective of RL agents, the environment they interact with can be represented as a state-transition graph, which models transitions between possible states within a Markov Decision Process (MDP). In this context, the value-updating process (illustrated in Figure 1 (1)) involves iteratively propagating value information across known states on a sampled state-transition graph (empirical state-transition graph) until training concludes. However, for environments with large state-transition graphs, this process becomes computationally expensive. The value of each state must be propagated back state by state from all potential future states, leading to significant delays in convergence.

A common approach to address this issue is to incorporate model-based planning algorithms, such as Monte Carlo Tree Search (MCTS) (Schrittwieser et al., 2020) and Value Iteration (VI) (Tamar et al., 2016), to estimate and propagate future state values. However, MCTS can be slow for action selection as it requires extensive sampling at decision time (Swiechowski et al., 2023). This limitation has led us to focus on the potential of the classical VI algorithm for improving training efficiency.

The primary drawback of VI lies in its requirement to iterate over a large tensor with dimensions $|\mathcal{S}| \times |\mathcal{A}| \times |\mathcal{S}|$, where $\mathcal{S}$ and $\mathcal{A}$ denote the state and action spaces, respectively. During each iteration, the value of state $s_t$ is only propagated back incrementally to update the preceding state $s_{t-1}$, limiting efficiency.

In this paper, we present an enhancement to the VI algorithm by introducing a novel graph structure to accelerate the value-updating process in RL. Our approach draws inspiration from real-world highway systems, where drivers traverse significant, frequently visited destinations without the need for constant decision-making at intersections, enabling high-speed travel along unbranched paths. Similarly, we identify paths without branching within the empirical state-transition graph, which we term *highways*. Along these highways, value updates for states can be aggregated and performed as a single operation, as opposed to incrementally updating values state by state. By enabling efficient propagation of values over long ranges from potential future states simultaneously, our method significantly enhances overall training efficiency. This notion of multi-step efficiency has also shown promise in the context of representation learning in RL (McInroe et al., 2024).

We introduce the concept of a *highway graph* for modeling the environment, shown in Figure 1. In this work, we focus on deterministic environments with discrete state and action spaces. The highway graph is constructed from the empirical state-transitions graph (Figure 1 (1)). In the highway graph, nodes are intersection states, and edges are highways. A highway is formed using the sequence of transitions in a non-branching path that only bridges two states without adjacent (Figure 1 (2)). Decisions are only necessary at

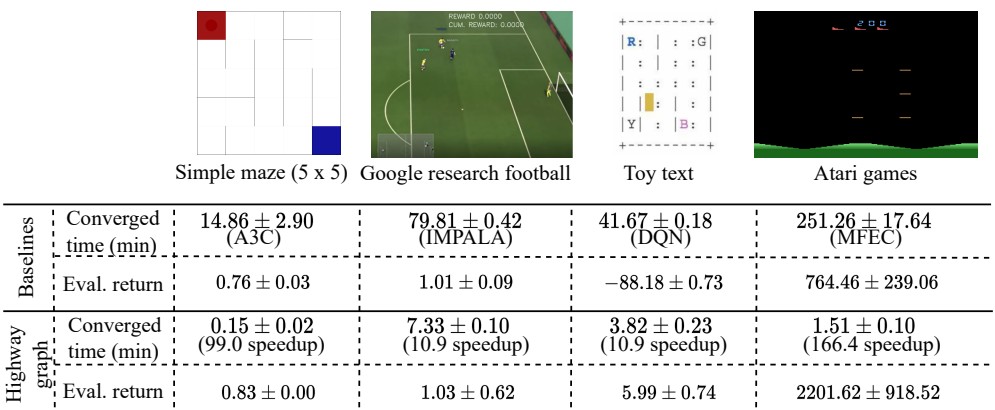

| | | Simple maze (5 x 5) | Google research football | Toy text | Atari games |
|---|---|---|---|---|---|
| Baselines | Converged time (min) | $14.86 \pm 2.90$ (A3C) | $79.81 \pm 0.42$ (IMPALA) | $41.67 \pm 0.18$ (DQN) | $251.26 \pm 17.64$ (MFEC) |
| | Eval. return | $0.76 \pm 0.03$ | $1.01 \pm 0.09$ | $-88.18 \pm 0.73$ | $764.46 \pm 239.06$ |
| Highway graph | Converged time (min) | $0.15 \pm 0.02$ (99.0 speedup) | $7.33 \pm 0.10$ (10.9 speedup) | $3.82 \pm 0.23$ (10.9 speedup) | $1.51 \pm 0.10$ (166.4 speedup) |
| | Eval. return | $0.83 \pm 0.00$ | $1.03 \pm 0.62$ | $5.99 \pm 0.74$ | $2201.62 \pm 918.52$ |

Figure 2: A comparison of converged time of training (within one million frames) and corresponding speedups by the highway graph compared to baselines. The first row of images are example states from each environment. The results demonstrate a 10 to more than 150 times faster RL agent training compared to baselines when adopting the highway graph. All the experiments were performed on the same machine with a 12-core CPU and 128 GB Memory.

intersection states, such as when a driver must choose a direction at an intersection, and continue along the highway otherwise. It worth noting that the concept of the highway graph is not equivalent to the concept of state abstraction (Chen et al., 2024). When the environment deviates from what is recorded in the highway graph, the agent can fall off the highway, gaining new experiences and incrementally expanding the highway. Thus, our approach does not reduce the problem's size or complexity.

Due to the highways, the original empirical state-transition graph shrinks into a very small highway graph, with two advantages for value learning. First, the values of any state connected by the highways are updated with a single operation, achieving a very high training efficiency. Second, no experience sampling for the value updating is required, and the VI algorithm can be conducted on the entire graph to avoid sub-optimal value learning caused by the sampling. We theoretically demonstrate the convergence to the optimal state value function of the VI algorithm on a highway graph.

As a model-based RL method, the highway graph still needs to be stored in memory. There is a possibility that the highway graph will be too large to fit in memory. To address this issue, we further replace the highway graph with a neural network-based agent obtained by re-parameterizing the value information of the highway graph.

We evaluate our proposed methods against a range of established and state-of-the-art RL approaches (Schrittwieser et al., 2020; Antonoglou et al., 2022; Danihelka et al., 2022; Lin et al., 2018; Hu et al., 2021; Blundell et al., 2016; Pritzel et al., 2017a; Mnih et al., 2015; Kapturowski et al., 2019; Schulman et al., 2017; Espeholt et al., 2018; Mnih et al., 2016) across four distinct categories of tasks. Figure 2 illustrates the speedups achieved by our highway graph approach compared to popular baselines across four different environments. Our highway graph consistently delivers on-par or superior expected returns, with speedups ranging from approximately 10-fold to over 150-fold compared to baselines, providing compelling evidence of the feasibility and effectiveness of converting to a highway graph structure. Notably, many tasks in the evaluated environments can be solved within just one million frames, a milestone that most baseline methods fail to achieve. Furthermore, the re-parameterized neural network, employed as the agent trained by the highway graph, delivers an additional 5% to 20% improvement in expected returns, attributed to its enhanced generalization capabilities.

In summary, our paper makes the following key contributions:

- We propose a novel graph structure, the highway graph, which facilitates more efficient value learning by enabling long-range value propagation within each learning iteration.

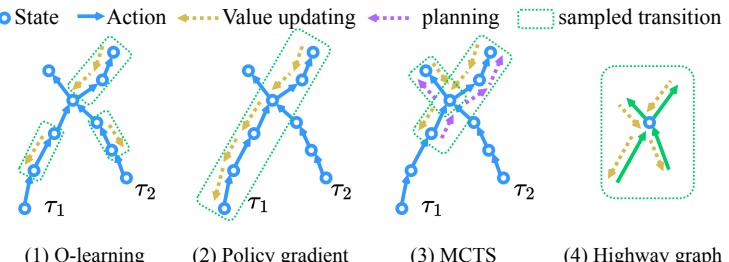

Figure 3: Types of value updating.

- We improve the training efficiency of the classic VI algorithm by leveraging the highway graph. Additionally, we provide a theoretical proof of convergence to the optimal state value function for the improved VI algorithm.

- We re-parameterize the highway graph into a neural network, resulting in a policy with superior generalization capabilities and reduced storage costs.

- We evaluate our framework on four categories of learning tasks, benchmarking it against state-of-the-art RL baselines. Our approach significantly reduces training time and increases sample efficiency, achieving speedups ranging from approximately 10-fold to over 150-fold compared to baselines. Furthermore, it solves tasks within one million frames while maintaining equal or superior expected returns.

## 2 Related work

In this section, we discuss how model-based and model-free RL methods learn state values, highlighting the differences between these approaches and our proposed highway graph method.

Model-based RL agents generally leverage the environment model—represented as a state-transition graph—through two primary approaches: value search and graph-based value iteration. Both approaches rely on the model to facilitate value learning, which aims to propagate values from terminal states back to the starting states within the state-transition graph.

*Value search* methods use tree search algorithms (Schrittwieser et al., 2020; Kaiser et al., 2020; Danihelka et al., 2022), such as MCTS, for both agent training and action selection through simulation. This process is often time-consuming. In value search, state values for action selection—referred to as planning—are typically determined within a specific range of the state-transition graph. The values of the leaf states in the search tree are usually provided by the state value function. The planning replies on sampling strategies to estimate the value of the current state, which can decrease the efficiency of the training, as illustrated in Figure 3 (3).

*Value iteration* methods obtain the state values by retrieving from a previously maintained memory of the state-transition model (Tamar et al., 2016). Also, episodic control algorithms (Blundell et al., 2016; Zhu et al., 2020; Pritzel et al., 2017a; Hu et al., 2021; Zhu et al., 2020) are in this category. The episodic control algorithms maintain a memory to make decisions that explicitly store the transition information as a memory. During value updating, the maximum value of either all the future states (Blundell et al., 2016) or states of related trajectories (Zhu et al., 2020) will be used to update the current state's value (Zhu et al., 2020). Such memory can also be parameterized to improve the generalization and performance (Lin et al., 2018; Pritzel et al., 2017b) which gives insight into the parameterized agent trained by memorized data.

Model-free RL algorithms do not explicitly learn the environment model but instead use sampled transitions from the replay to recognize the dynamics of the environment when learning the state value. The value-based and policy-based approaches are typical model-free RL algorithms.

*Q-learning* methods are typical value-based methods, like DQN (Mnih et al., 2015), which propagate the value of the next few states in the randomly sampled transitions to the current one in each update using $n$-step temporal difference error, see Figure 3 (1). However, not all valuable transitions are sampled for value updates. To solve this, a prioritized replay buffer was proposed to sample more useful transitions when training the agent (Schaul et al., 2016). Highway value iteration networks (Wang et al., 2024) optimize the $n$-step value updating process via a modified multi-step Bellman operator to avoid the value underestimation issue and make the value updating converge to the optimal value function. R2D2 improved DQN by using a recurrent encoder to obtain better state representations considering all past states in the episode (Kapturowski et al., 2019). The recurrent model considers the causal effect of the historical transition which gives us an insight into designing the encoder for the state.

*Policy gradient* methods update the value of states directly from the gradient of future value estimation from the sampled trajectories (Schulman et al., 2017; Che et al., 2023; Fatkhullin et al., 2023; Mnih et al., 2016), see Figure 3 (2). The training efficiency can be improved by using distributed systems (Mnih et al., 2016; Espeholt et al., 2018).

During the value update of the model-free agents, inconsistent value updates may occur. The value updating of these methods heavily depends on the sampled transitions or trajectories, and if transitions from $s_i$ with the highest value are not sampled, the updated value of $s_i$ will bias to the real value. Often more sampling is required to overcome this bias, which leads to low training efficiency. In addition, it also makes model-free RL training inefficient since the value of terminal states is propagated to the starting states state-by-state.

Consequently, there are major differences between the highway graph and the above methods, see Figure 3 (4). First, the highway graph itself acts as a memory to store the transiting information, and during value updating, the value will be propagated throughout the entire graph, leading to complete value propagation on each learning iteration. Second, the value updating operation on the highway graph is only for the states of intersection (much fewer than the original states) and highways, instead of all states and transitions, to reduce unnecessary computation and therefore increase training efficiency.

Many approaches also attempt to reduce the costs of value learning by task division. Hierarchical reinforcement learning methods learn the sub-goals of the environment hierarchically to reduce the problem-solving complexity (Nachum et al., 2018). Similarly, state abstraction methods, such as macro-action methods (van den Berg et al., 2012), use fixed action sequences to interact with environments for a smaller problem size. These methods are different from our method, since our highway RL agent directly interacts with the environment at every step without state abstraction or policy hierarchies, and is capable of exiting the highway at any time, thereby gaining new experiences and incrementally expanding the highway graph.

Improving the sample efficiency of value learning is important to speed up agent training. Offline data can be used to improve the sample efficiency of agent training by pre-training a policy before online RL training (Andres et al., 2025). Auxiliary learning targets can also help the agent to learn more effective representations (He et al., 2022; Liu et al., 2021) and enhance generalization to unseen environments (Dunion et al., 2023b;a; Yu et al., 2024). These methods share the same goal of our method but with different approaches.

## 3 Methodology

In this section, we will introduce a novel graph structure to model the MDP of the environment, named highway graph. The highway graph is built upon the empirical state-transition graph to accelerate value updating, as well as action selection for the environment, see Figure 4 for the overall data flow of our highway graph RL method.

### 3.1 Markov decision process and empirical state-transition graph

Generally, RL agents try to maximize the expected return when interacting with the environment (Kaelbling et al., 1996). The environment can be described as a Markov decision process (MDP) (Sutton & Barto, 2018) by a 5-tuple $(\mathcal{S}, \mathcal{A}, \mathcal{T}, \mathcal{R}, \gamma)$, where $\mathcal{S}$ and $\mathcal{A}$ denote the spaces for state and action, $\mathcal{T}(s', a, s) = p(s'|a, s)$ is the probability of transition into state $s'$ given current state $s$ and action $a$, $\mathcal{R}(s, a)$ is the immediate reward

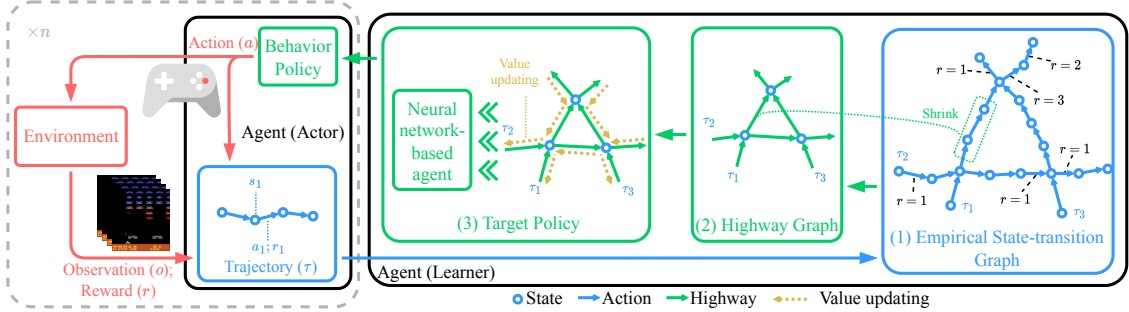

Figure 4: Overall data flow of our highway graph RL method. The actor (on the left) sends the sampled transitions by the behavior policy to the learner (on the right) which (1) constructs the empirical state-transition graph with rewards (in Section 3.1); (2) converts the empirical state-transition graph to the corresponding highway graph (Section 3.2); (3) updates the value of state-actions in the highway graph by an improved value iteration algorithm and re-parameterize the highway graph to a neural network-based agent as the new behavior policy (Section 3.3).

of action $a$ in current state $s$, and $0 \leqslant \gamma < 1$ is the discount factor. With known transition and reward function, the state value function of the policy at learning step $u$ can be obtained by value update as follows,

$$V^{(u)}(s) = \max_a \left[ \mathcal{R}(s, a) + \gamma \sum_{s'} \mathcal{T}(s', a, s) V^{(u-1)}(s') \right], \tag{1}$$

where $V^{(0)}(s)$ can be initialized as 0. For deterministic environments, $\mathcal{T}(s', a, s)$ is constant and $\sum_{s'} \mathcal{T}(s', a, s) V^{(u-1)}(s')$ will be $\mathcal{T}(s', a, s) V^{(u-1)}(s')$.

**Empirical transition and reward functions**   For general RL methods, obtaining the transition and reward functions of the entire MDP is merely impossible. Only empirical state transitions and rewards, sampled from the environments, are used by the RL algorithms. Therefore, we provide a method that efficiently updates state values for the empirical transition and reward functions. The empirical transition and reward functions are defined as

**Definition 1** (Empirical Transition and Reward Functions). *Given an MDP $(\mathcal{S}, \mathcal{A}, \mathcal{T}, \mathcal{R}, \gamma)$, assume that we sampled $k$ samples $(s'_i, a_i, s_i, r_i)$ with some policy $\pi$, then the empirical transition function $\hat{\mathcal{T}}$ is*

$$\hat{\mathcal{T}}(s', a, s) = \frac{\sum_{i=1}^{k} \mathbb{1}(s' = s'_i, a = a_i, s_i = s)}{\sum_{i=1}^{k} \mathbb{1}(a = a_i, s_i = s)}, \tag{2}$$

*and many consecutive transitions in the $k$ samples form the trajectory $l \in \mathcal{L}$ which is*

$$l = \bigcup_{i=m}^{n} \{(s'_i, a_i, s_i, r_i)\}, \tag{3}$$

*where $m$ and $n$ is the beginning and terminal time step of $l$. The empirical reward function $\hat{\mathcal{R}}$ is*

$$\hat{\mathcal{R}}(s, a) = \begin{cases} r_i, & \text{if} \quad \exists i \quad \text{s.t.} \quad s = s_i, a = a_i, \\ 0, & \text{otherwise.} \end{cases} \tag{4}$$

**Empirical State-transition graph**   The empirical state-transition graph is from the empirical transition and reward functions which can be defined as

**Definition 2** (Empirical State-Transition Graph). *Given an empirical transition function $\hat{\mathcal{T}}$ and an empirical reward function $\hat{\mathcal{R}}$, the corresponding empirical state-transition graph is given by $\mathcal{G}_{\hat{\mathcal{T}}} = \left\{ \mathcal{S}, \mathcal{E}_{\hat{\mathcal{T}}} \right\}$, where $\mathcal{S}$ is the node set (i.e. each node corresponds to a state), and the edge $(s', a, s, \hat{\mathcal{R}}(s, a)) \in \mathcal{E}_{\hat{\mathcal{T}}}$ if and only if $\hat{\mathcal{T}}(s', a, s) > 0$.*

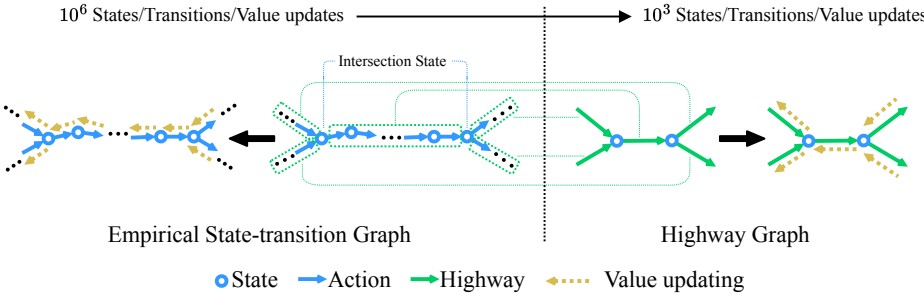

Figure 5: Intuitive idea of the highway graphs. The path without branching in the empirical state-transition graph can be merged as a highway. Value updating on the highway graph will be much less computationally extensive due to the dramatic reduction of the nodes and edges.

In a deterministic environment, an action $a_i$ given current state $s_i$ uniquely determines the consequent state $s_j$. Therefore, $\hat{\mathcal{T}}(s', a, s) \in \{0, 1\}$ in deterministic environments.

### 3.2 Highway graph

In our work, we convert the empirical state-transition graph into the highway graph with reduced states and transitions, shown in Figure 4 (2). What is a highway graph and how to obtain it are discussed in Section 3.2.1 and Section 3.2.2 respectively.

#### 3.2.1 Highway and highway graph

Naively, the value iteration algorithm requires iterating over a space of the whole empirical state-transition graph ($\mathcal{S} \times \mathcal{A} \times \mathcal{S}$) to reach the converged state value function, leading to a very high computational cost (van den Berg et al., 2012). To tackle this problem, we propose a new graph structure, named highway graph, which is obtained from the original empirical state-transition graph.

Instead of naively iterating over the whole empirical state-transition graph, we seek to merge various value updates to reduce the computations. The intuitive idea of the highway graph is to merge the non-intersection states in the empirical state-transition graph, which converts a path in between intersection states, into a highway (an edge of the highway graph) to reduce the overall graph size, see Figure 5. The intersection states (nodes in the highway graph), which fork, merge, or loop state transitions, remain the same in the highway graph. Within this smaller, yet equivalent, highway graph, we can define a new value update that effectively merges a few value updates under the original empirical state transition graph. This leads to better efficiency since the highway graph is smaller, which means fewer value updates are required.

The edge from the merging, i.e. highway, denoted as $h \in \mathcal{H}$, is defined in Definition 3.

**Definition 3** (Highway). *Given an empirical state-transition graph $\mathcal{G}_{\hat{\mathcal{T}}} = \left\{\mathcal{S}, \mathcal{E}_{\hat{\mathcal{T}}}\right\}$ that are constructed from a series of samples $(s_i', a_i, s_i, r_i)$, if for some $s, s' \in \mathcal{S}$, there is only one trajectory $l$ from $s$ to $s'$ in the empirical state-transition graph, then $l$ between $s$ and $s'$ will be considered as a highway, denoted as $h_{s,s'} \in \mathcal{H}$:*

$$h_{s,s'} = l_{s,s'}. \tag{5}$$

Based on highways, the highway graph with shrunk graph size is defined in Definition 4. The merging to make the highways occurs for all possible nodes between non-branching paths, and only intersection nodes will be excluded. Thus, the highway graph is the minimal graph structure to fulfill identical value updating on the original empirical state-transition graph. The value propagation of state transitions will be replaced by a single operation on highways as much as possible, and the value updating process will consequently speed up. Practical examples of the highway graph can be found in Section 4.2 and Section 4.3.

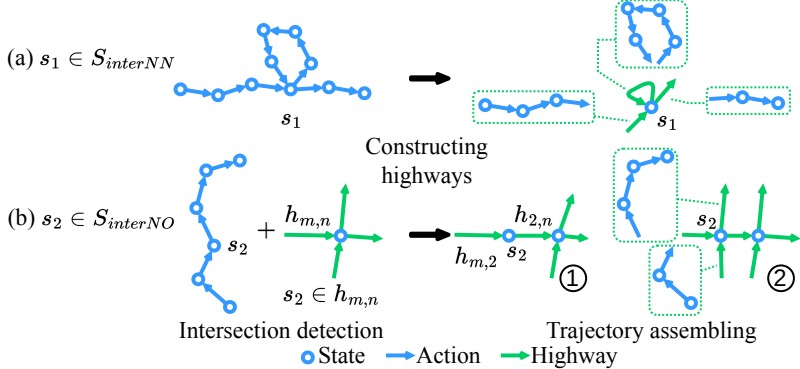

Figure 6: The highway graph construction.

**Definition 4** (The highway graph). *Given an empirical state-transition graph $\mathcal{G}_{\hat{\mathcal{T}}} = \left\{ \mathcal{S}, \mathcal{E}_{\hat{\mathcal{T}}} \right\}$ that are constructed from a series of samples $(s_i', a_i, s_i, r_i)$, its corresponding highway graph $\mathcal{G}_{\mathcal{H}} = \{\mathcal{S}_{inter}, \mathcal{H}\}$ can be constructed as follows:*

(1) *$\mathcal{S}_{inter} = \{s | \exists s_i', s_j' \in \mathcal{S}, s_i' \neq s_j' \neq s$, such that there are transitions from $s$ to $s_i'$ ($\hat{\mathcal{T}}(s_i', a, s) > 0$) and from $s$ to $s_j'$ ($\hat{\mathcal{T}}(s_j', a, s) > 0$) in $\mathcal{E}_{\hat{\mathcal{T}}}$, or there are transitions from $s_i'$ to $s$ ($\hat{\mathcal{T}}(s, a, s_i') > 0$) and from $s_j'$ to $s$ ($\hat{\mathcal{T}}(s, a, s_j') > 0$) in $\mathcal{E}_{\hat{\mathcal{T}}}\}$;*

(2) *$\forall s, s' \in \mathcal{S}_{inter}, h_{s,s'} \in \mathcal{H}$ if and only if there exist only one trajectory of $\mathcal{G}_{\hat{\mathcal{T}}}$ starts from $s$ to $s'$.*

### 3.2.2 Highway graph construction

Based on this idea, we next give the incremental algorithm to continually construct the highway graph from the sampled trajectory $l$. The entire algorithm can be divided into two phases: intersection detection and trajectory assembling, shown in Figure 6.

**Intersection detection** It aims to find all the intersection states and store them into $\mathcal{S}_{inter}$, which is initially an empty set, for the nodes of the highway graph. There are mainly two sources of intersection states. (1) it happens among new states of the newly sampled trajectory by revisiting previous states in the same sampled trajectory, denoted as $\mathcal{S}_{interNN}$; (2) or during the merge between the new trajectories and the existing old highway graph, denoted as $\mathcal{S}_{interNO}$.

The first type of the intersection state is given by Equation 6.

$$
\begin{aligned}
\mathcal{S}_{interNN} = &\left\{ s_t : s_t, s_{t+1} \in l, s_t \in \mathcal{S}_{visited}, s_{t+1} \notin \mathcal{S}_{visited} \right\} \cup \\
&\left\{ s_{t+1} : s_t, s_{t+1} \in l, s_t \notin \mathcal{S}_{visited}, s_{t+1} \in \mathcal{S}_{visited} \right\} \cup \\
&\left\{ s_t, s_{t+1} : s_t, s_{t+1} \in l, s_t, s_{t+1} \in \mathcal{S}_{visited}, (s_{t+1}, a_t, s_t) \notin \mathcal{T}_{visited} \right\},
\end{aligned}
\tag{6}
$$

where $\mathcal{S}_{visited} = \cup_{k=0}^{t-1} s_k$ and $\mathcal{T}_{visited} = \cup_{k=0}^{t-1} (s_{k+1}, a_k, s_k)$ are the visited states and transitions of the current step $t$ in the trajectory $l$. Sets in different lines of Equation 6 are for the forking, merging, and crossing of the current trajectory respectively.

Another type of the intersection state is given by Equation 7.

$$
\begin{aligned}
\mathcal{S}_{interNO} = &\left\{ s_t : s_t, s_{t+1} \in l, s_t \in \mathcal{G}_{\mathcal{H}}, s_{t+1} \notin \mathcal{G}_{\mathcal{H}} \right\} \cup \\
&\left\{ s_{t+1} : s_t, s_{t+1} \in l, s_t \notin \mathcal{G}_{\mathcal{H}}, s_{t+1} \in \mathcal{G}_{\mathcal{H}} \right\} \cup \\
&\left\{ s_t, s_{t+1} : s_t, s_{t+1} \in l; s_t, s_{t+1} \in \mathcal{G}_{\mathcal{H}}, \hat{\mathcal{T}}(t+1, a, t) = 0 \right\},
\end{aligned}
\tag{7}
$$

---

**Algorithm 1** Highway graph incremental construction

---

**Input**   : New trajectories $\mathcal{L}$ and existing $\mathcal{G}_{\mathcal{H}}$
**Output:** The updated highway graph $\mathcal{G}_{\mathcal{H}}$

**1 foreach** $l \in \mathcal{L}$ **do**
**2**  |  Let source state $s_i$ be $s_0 \in l_0 = (s'_0, a_0, s_0, r_0)$
**3**  |  Find all intersection states $\mathcal{S}_{interNN} \cup \mathcal{S}_{interNO}$ of $l$ using Equation 6 and Equation 7
**4**  |  **foreach**  *current intersection state* $s_j \in sort\left(\{s_j : s_j \in \mathcal{S}_{interNN} \cup \mathcal{S}_{interNO}\}\right)$ **do**
**5**  |  |  Search sub-trajectory $l_{t:t'}$ that bridge $s_i$ and $s_j$
**6**  |  |  Convert $l_{t:t'}$ into highway $h_{i,j}$ according to Definition 3
**7**  |  |  **if** $s_j \in \mathcal{S}_{inter}$ **then**
**8**  |  |  |  **if** $h_{i,j} \notin \mathcal{H}$ **then**
**9**  |  |  |  |  Let $\mathcal{H}$ be $\mathcal{H} \cup h_{i,j}$
**10** |  |  **else if** $s_j \in h_{m,n}$ where $h_{m,n} \in \mathcal{H}$ **then**
**11** |  |  |  Let $\mathcal{S}_{inter}$ be $\mathcal{S}_{inter} \cup s_j$
**12** |  |  |  Split $h_{m,n}$ into $h_{m,j}$ and $h_{j,n}$
**13** |  |  |  Let $\mathcal{H}$ be $\mathcal{H} \cup \{h_{m,j}, h_{j,n}\} \setminus h_{m,n}$
**14** |  |  **else**
**15** |  |  |  Let $\mathcal{S}_{inter}$ be $\mathcal{S}_{inter} \cup s_j$
**16** |  |  |  Let $\mathcal{H}$ be $\mathcal{H} \cup h_{i,j}$
**17** |  |  Let source state $s_i$ be $s_j$
**18** |  Let $\mathcal{H}$ be $\mathcal{H} \cup h_{j,T}$ where $|l| = T$
**19 return** $\mathcal{G}_{\mathcal{H}} = \{\mathcal{S}_{inter}, \mathcal{H}\}$

---

where $s_t \in \mathcal{G}_{\mathcal{H}}$ means either $s_t \in \mathcal{S}_{inter}$ or $s_t \in h (h \in \mathcal{H})$, and $\hat{\mathcal{T}}(t+1, a, t)$ is the transition function for any action $a$ from state $s_t$ to $s_{t+1}$ of $\mathcal{G}_{\mathcal{H}}$. The first two sets of Equation 7 indicate whether $s_t$ or $s_{t+1}$ of current trajectory $l$ is in the existing highway graph $\mathcal{G}_{\mathcal{H}}$ (either is the existing intersection or within a highway $h \in \mathcal{H}$) that should be considered as possible new intersection nodes. The last set of Equation 7 is for the new intersections in an existing highway or is the existing intersections in the $\mathcal{G}_{\mathcal{H}}$. Thus, the intersection states will be in Equation 8.

$$\mathcal{S}_{inter} = \mathcal{S}_{inter} \cup \{s_i : s_i \in \mathcal{S}_{interNN} \cup \mathcal{S}_{interNO}\} \tag{8}$$

**Trajectory assembling**   This phase will split the new trajectory $l$, by the newly found intersection states, and convert them into highways. If the intersection state is on the existing highway, this highway will be split into two highways by this state. After that, the intersection states and highways will be used to assemble the highway graph. Algorithm 1 shows the process of highway graph construction. The function $sort()$ sorts the input set of states in ascending order according to the time step of the trajectory.

After the construction of the highway graph, the original MDP can be converted to a new MDP, which we call the highway MDP, defined in Definition 5. The transition and reward function of the highway MDP is different from the ones in the original MDP. The transition function of highway MDP is only for intersection states in the highway graph, and the reward function is for the aggregated reward of highways instead of individual states. The action in the highway graph for $s_i$, which transits the agent to $h_{i,j}$, will bring the agent from $s_i$ to $s_j$.

**Definition 5** (Highway MDP). *The highway graph $\mathcal{G}_{\mathcal{H}}$ convert the classic MDP $(\mathcal{S}, \mathcal{A}, \mathcal{T}, \mathcal{R}, \gamma)$ into a highway MDP, denoted as $MDP_{\mathcal{H}} = \left(\mathcal{S}_{inter}, \mathcal{A}, \mathcal{T}_{\mathcal{H}}, \mathcal{R}_{\mathcal{H}}, \gamma\right)$. $\mathcal{R}_{\mathcal{H}} = \mathcal{R}\left(\mathcal{S}_{inter}, \mathcal{A}\right)$ and*

$$\mathcal{T}_{\mathcal{H}}(s_i, a, s_j)_{s_i, s_j \in \mathcal{S}_{inter}} = \begin{cases} 1, & h_{i,j} \in \mathcal{H} \text{ and } a \text{ is the first action of } h_{i,j}, \\ 0, & \text{otherwise}. \end{cases} \tag{9}$$

---

**Algorithm 2** Value updating on highway graph

---

    **Input** : Highway graph $\mathcal{G}_h = \{\mathcal{S}_{inter}, \mathcal{H}\}$, max learning iteration $U$, threshold of changes $\delta$
    **Output:** Highway graph with estimated state values $\mathcal{G}'_{\mathcal{H}}$

**1** Let $u$ be 0
**2** **while** $u < U$ **do**
**3**    Update state value function $V_{\mathcal{H}}^{(u)}(s_i)$ using Equation 11
**4**    Update state-action value function $Q_{\mathcal{H}}^{(u)}(s_i, a_j)$ using Equation 12
**5**    Set $u$ be $u + 1$
**6**    **if** $\sum_{s_i \in \mathcal{S}_{inter}} \sum_{a_j \in \mathcal{A}} \Delta Q_{\mathcal{H}}^{(u)}(s_i, a_j) < \delta$ **then**
**7**       | break

---

### 3.3 Value updating and target policy

During value updating, we introduce the graph Bellman operator as the planning module to iterate values on the highway graph. After that, we discuss how highway graphs help to speed up value updating and act as the policy afterward to tackle the task in the environment, as shown in Figure 4 (3).

#### 3.3.1 Value updating on the highway graph

Once the highway graph and its highway MDP are obtained, a more efficient yet accurate value update of all states can be performed based on value iteration.

Highway MDP shares the same properties as the classic MDP, and the state values during updating are still in a complete metric space shown by Lemma 1, and RL algorithms suitable for MDP can be applied to Highway MDP as well. Lemma 1, associated with Lemma 2, will be used to show the convergence of value updating on the highway, shown in Proposition 1.

**Lemma 1.** *Denote the state value vector* $v = \left[\hat{V}(s_1), \hat{V}(s_2), \cdots, \hat{V}(s_i), \cdots\right]^{\mathsf{T}} \in \mathcal{V}$, *where* $\mathcal{V} \in \mathbb{R}^{|\mathcal{S}_{inter}|}$, $\hat{V}$ *is an estimate of state value function* $V$, *and* $s_i \in \mathcal{S}_{inter}$ *is from Highway MDP, and define a metric* $d(w, v) = \|w - v\|_{\infty}$ $(w, v \in \mathcal{V})$, *then* $(\mathcal{V}, d)$ *is a complete metric space.*

Bellman equation is widely used by temporal-difference learning (Sutton & Barto, 2018). Given the currently sampled empirical transitions from the environment, states' values can be propagated within several time steps. However, this update process only considers the future value along with one trajectory per iteration and leads to non-optimal value updates if the best transitions are not sampled for the training. To avoid this issue, we give the graph Bellman operator to update the value of states in a value iteration scheme on the highway graph, considering all the known future states for accurate value updating, defined in Definition 6.

**Definition 6** (Graph Bellman operator). *Denote* $\Gamma_i^1$ *is the 1-order neighbors of* $s_i$ *on the highway graph* $\mathcal{G}_{\mathcal{H}}$, *i.e.* $h_{i,j} \in \mathcal{H}$ *for* $s_j \in \Gamma_i^1$, *and a max pooling function* $\Phi$ *to retrieve the effective value propagated from the neighbor. Given the complete metric space* $(\mathcal{V}, d)$, *the graph-based Bellman operator is* $G(V(\mathcal{S}_{inter})) : \mathcal{V} \to \mathcal{V}$, *and* $G_i$ *is*

$$G_i(V(s_i)) = \Phi_{s_j \in \Gamma_i^1} \left(\mathcal{R}(s_i, a) + \gamma V(s_j)\right), \tag{10}$$

*where* $s_i \in \mathcal{S}_{inter}$.

Graph Bellman operator is still a contracting mapping given in Lemma 2.

**Lemma 2.** *Graph Bellman operator* $G$ *is a max-norm* $\gamma$-*contraction mapping.*

According to the graph Bellman operator, the $u$-th value updating on the highway graph to obtain the state value and state-action value functions will correspondingly be

$$V_{\mathcal{H}}^{(u)}(s_i) = \Phi_{s_j \in \Gamma_i^1}(V_{\mathcal{H}}^{(u-1)}(s_j) \cdot \gamma^{|h_{i,j}|} + r_{h_{i,j}}), \tag{11}$$

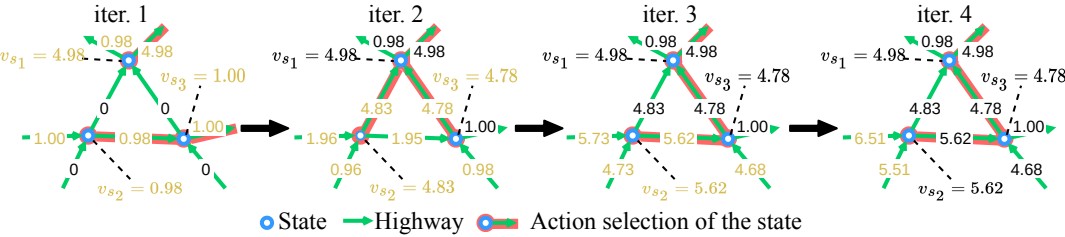

Figure 7: Value updating and action selection of corresponding highway graph of the empirical state-transition graph in Figure 4 (1). The discount factor $\gamma = 0.99$. The updated values of states and highways are in yellow in each learning iteration. The selected actions of intersection states are changing according to the latest learned value.

$$Q_{\mathcal{H}}^{(u)}(s_i, a_j) = V_{\mathcal{H}}^{(u-1)}(s_j) \cdot \gamma^{|h_{i,j}|} + r_{h_{i,j}}, \tag{12}$$

where $s_i, s_j \in \mathcal{S}_{inter}$, and $a_j$ is the action to $s_j$ from $s_i$. $V_{\mathcal{H}}^{(0)}(s_i)$ and $Q_{\mathcal{H}}^{(0)}(s_i, a_j)$ are initialized with zeros. The reward of the highway $r_{h_{i,j}}$ is

$$r_{h_{i,j}} = \sum_{t=1}^{|h_{i,j}|} r_t \cdot \gamma^t, \tag{13}$$

where $r_t$ is the reward of the $t$-th state $s_t$ in the highway. The value updating iterations of the empirical state-transition graph in Figure 4 (1) are shown in Figure 7, and the value of states are obtained by Algorithm 2. The process calculates the value of both intersection states and highways for $U$ (less than $\infty$) rounds until the state value function reaches optimal.

Next, we analyze the convergence of highway graph value updating in Proposition 1, and we further give the speed advantages of value updating using the highway graph.

**Proposition 1** (Convergence of value updating on highway graphs.). *Let $(\mathcal{V}, d)$ be a complete metric space of the $MDP_{\mathcal{H}} = \left( \mathcal{S}_{inter}, \mathcal{A}_{\mathcal{H}}, \mathcal{T}_{\mathcal{H}}, \mathcal{R}_{\mathcal{H}}, \gamma \right)$, sufficient application of the graph Bellman operator $G$ for value updating on highway graph, according to Equation 11, converges to a unique optimal state value function $V_{\mathcal{H}}^*$.*

**Remark 1.** *The advantage of the highway graph is that it uses a much smaller highway graph to replace the original empirical state-transition graph. This replacement leads to high computational speedup. The time complexity of value iteration on the original empirical state-transition graph is $\mathcal{O}\left(|\mathcal{S}| \cdot |\mathcal{A}| \cdot |\mathcal{S}|\right)$ where $|\mathcal{S}|$ and $|\mathcal{A}|$ are the numbers of states and actions of the empirical state-transition graph, respectively. Highways merge trajectories without branching, and the ratio of reduced states to original states is $z = \frac{|\mathcal{S}_{inter}|}{|\mathcal{S}|}$. Consequently, the time complexity of value updating on the highway graph is $\mathcal{O}\left(|\mathcal{S}_{inter}| \cdot |\mathcal{A}| \cdot |\mathcal{S}_{inter}|\right)$, which is $\frac{1}{z^2}$ times smaller than that of vanilla value iteration algorithm.*

### 3.3.2 Highway graph as the policy

Once the values of states are obtained, the highway graph will be used as the policy. An example of action selection by the highway graph is also shown in Figure 7. The selected action for the state may change during the value updating until the highway graph with optimal state values is obtained. The action will be accordingly generated by the policy in Equation 14 to interact with the environment.

$$a_{s_i} = \begin{cases} \arg\max_a Q(s_i, a) & s_i \in \mathcal{S}_{inter} \\ a_{h_{i,j}} & s_i \in h_{i,j} \\ \mathrm{random}(s_i) & s_i \notin \mathcal{G}_{\mathcal{H}} \end{cases}, \tag{14}$$

where $a_{h_{i,j}}$ is the action recorded in the highway $h_{i,j}$, and $\mathrm{random}(s_*)$ generate the action $a$ for $s_*$ by uniformed random sampling from the valid actions $\mathcal{A}$.

### 3.3.3   Highway graph re-parameterization

Despite the compact nature of value updating, an important property of the highway graph is the increasing graph size during construction from newly sampled empirical transitions. Thus, it is required to use a neural network-based agent, derived from the highway graph, to interact with the environment. Ideally, a neural network-based agent can inherit useful state-action value information from the highway graph, and maintain the generalization ability of neural networks. To this end, we re-parameterize the state-action value of the highway graph to obtain a neural network-based agent to (1) reduce the storage cost for the growing highway graph; and (2) generalize the highway graph to more unknown states. By doing this the highway graph can be considered as a jump-starter for both model-free and model-based RL agent training at the early training stage, by providing the state transitions and its values.

Take the model-free Q-learning algorithm as an example, the highway graph can be re-parameterized as the initial state-action value function $Q_\theta(s_i, a)$ before continuing improvement using the Q-learning. The initial neural state-action value function $Q_\theta(s_i, a)$ can be trained using the state-action pairs with learned values from the highway graph through supervised learning, and a simple MSE loss between $Q_\mathcal{H}(s, a)$ and $Q_\theta(s, a)$ can be used during re-parameterization. Denote $Q_\theta$ as the trained neural network-based agent utilizing the highway graph. The action $a_{s_i}$ generated by $Q_\theta$ for the $s_i$ will be

$$a_{s_i} = \arg\max_a Q_\theta(s_i, a). \tag{15}$$

## 4   Experiments

In this section, we first evaluate the expected return (accumulated rewards with discount) versus frames and computed time respectively on different categories of environments to answer the following questions: (1) is our proposed highway graph RL method suitable for both simple and complex environments? (2) can our method converge with less sampled experience/compute time to a policy yielding higher returns compared to other baselines? Next, we analyze the functionality and behavior of our method by investigating: (3) evidence of the structural advantage of the highway graph; (4) effectiveness and efficiency details of value updating on the highway graph; (5) the performance of the re-parameterized neural network-based agent.

**Environments**   To better show the training efficiency advantages of our highway graph RL method, we only use one million frames of interaction from different types of Environments. Whether the information from one million frames is enough to solve the task in the environments will also be shown. Different random seeds are used to guarantee the deterministic assumption of the environments. Types of environments are listed below.

- Simple Maze [1]: a simple maze environment with customizable sizes.

- Toy Text (Towers et al., 2023): a tiny and simple game set, with small discrete state and action spaces, including FrozenLake, Taxi, CliffWalking, and Blackjack.

- Google Research Football (GRF) (Kurach et al., 2020): a physical-based football simulator.

- Atari learning environment (Bellemare et al., 2013): a simulator for Atari 2600 console games. These games are different, interesting, and designed to be a challenge for reinforcement learning.

**Baselines**   We compare our highway graph RL method (HG) with different baselines for different environments based on difficulty and complexity. For Simple Maze and Toy Text, our method is compared with model-free Q-learning methods DQN (Mnih et al., 2015) and policy gradient method A3C (Mnih et al., 2016) and PPO (Schulman et al., 2017), as well as model-based method MuZero (Schrittwieser et al., 2020) and its variants including Stochastic MuZero with a more powerful transition model (Antonoglou et al., 2022) and Gumbel MuZero with optimized root action selection for MCTS (Danihelka et al., 2022), to show how

---

[1] https://github.com/MattChanTK/gym-maze

Table 1: The HG behavior on The Simple Maze. We evaluate the convergence to the optimal state value function of value updating on the highway graph on three maze environments of varying sizes. We show the total reward and average discount return after convergence, frames and time taken until convergence, as well as the minimum number of learning iterations required for the convergence.

|  | Total Reward | Discounted avg. return | Frames used (M) | Wall clock time (min) | Converged learning iteration(s) |
|---|---|---|---|---|---|
| 3x3 | 0.97±0.01 | 0.93±0.01 | 0.07±0.01 | 0.03±0.01 | 1 |
| 5x5 | 0.95±0.01 | 0.82±0.02 | 0.09±0.01 | 0.12±0.02 | 1 |
| 15x15 | 0.96±0.01 | 0.37±0.05 | 0.43±0.02 | 1.01±0.04 | 1 |

different categories of RL methods learn the state before update its value (as shown in Figure 3) and the advantages of the highway graph. For more difficult and computationally demanding GRF environments, our method is compared with the Q-learning method R2D2 (Kapturowski et al., 2019) and policy gradient method IMPALA (Espeholt et al., 2018) to reveal the efficiency advantage during value updating among important model-free RL methods and our model-based highway graph RL method. The Atari games are important image-based environments for RL algorithm evaluation due to their versatility, so we take both classic methods including DQN, PPO, A3C, and state-of-the-art episodic-control-based methods including EMDQN (Lin et al., 2018), GEM (Hu et al., 2021), MFEC (Blundell et al., 2016), NEC (Pritzel et al., 2017a) as the baselines for a more comprehensive comparison. The reason we include the episodic-control-based methods is that these methods also use a memory module of transition to speed up the value updates which is similar to the highway graph RL method. In addition, we use RLlib (Liang et al., 2018) implementations for DQN, PPO, A3C, R2D2, and IMPALA. Other baselines including NEC, MFEC, EMDQN, GEM, and Gumbel MuZero are obtained from its official repository. The network architectures and hyperparameters of different baselines are given in the Appendix.

**Experiment setup** Our method also adopts an actor-learner training scheme similar to SEED RL (Espeholt et al., 2020). During the experiments, our highway graph RL agent is trained from scratch to compare to other baselines. The general training process is as follows. (1) The actors are guided by the highway graph to get the experiences in episodes with expected returns. Thus, initially, when the highway graph is empty, the actors randomly explore the environment before the first highway graph is constructed. The values of $\epsilon$ employed across the actors for $\epsilon$-greedy strategy are equally spaced, starting from 0.1 and ending at 1.0. During training, if the actors see an observation not in the highway graph, a random action is taken. (2) New experiences from the actors will be used to incrementally update the highway graph by the learner, and a value learning of the highway graph will be fulfilled for better action selection. During the value learning, our method propagates the values among nodes in the highway graph based on topological adjacency. The number of episodes used for incrementally highway graph updating vary across environments: 10 for Simple Maze and Atari games, 20 for Toy Text, and 50 for GRF. (1) and (2) are processed alternatively for iterations until the end of training. All the experiments were running in the Docker container with identical system resources including 8 CPU cores with 128 GB RAM, and an NVIDIA RTX 3090Ti GPU with 24 GB VRAM. The reported result of each experiment is from the averaged results among 10 different runs.

## 4.1 Comparison with baselines

### 4.1.1 Simple Maze

The Simple Maze environment generates a grid-like maze whose width and length can be specified. A sequence of adjacent grids not separated by a line constitutes a path, and the lines between grids are the walls that the agent cannot get through. The state of this environment at each step will be the current agent location, i.e. 2D coordinates, in the maze. The agent can move within the maze with up, down, left, and right actions. The RL agent needs to find a path from the blue/start point in the upper left to the red/terminate point in the lower right. Once the path is found, a reward of +1.0 can be obtained. A very

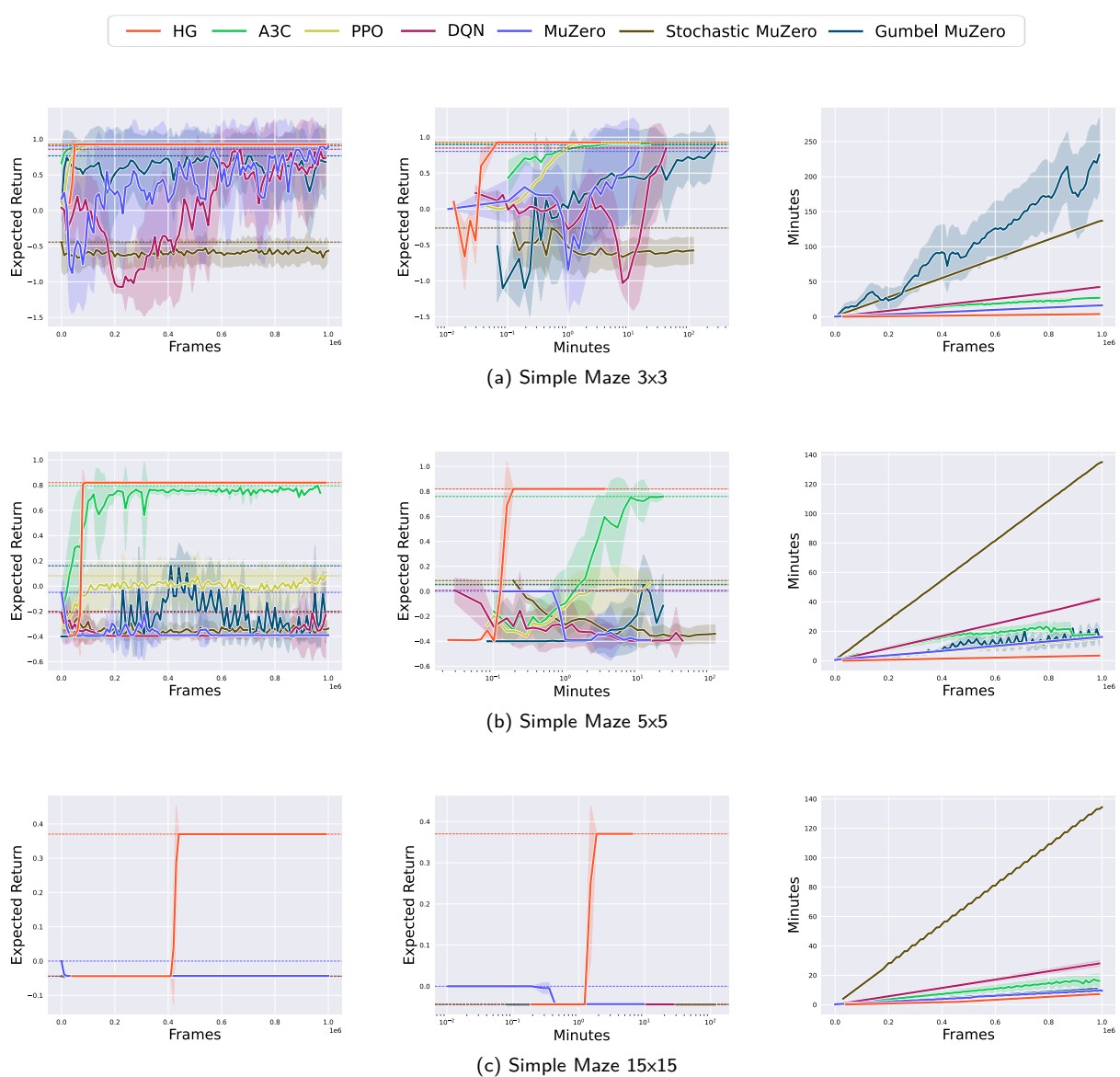

Figure 8: Expected return and efficiency comparison on 3x3, 5x5, and 15x15 Simple Maze environments. The lines and shaded areas are for the average values and the corresponding range ($\pm$) of standard deviations, and dashed horizontal lines show the asymptotic performance. Each run of every method was recorded from three perspectives simultaneously: frames and minutes versus expected return to demonstrate the performance over frames and time (in the first two columns), and frames versus minutes to illustrate the relative training efficiency (in the last column).

small penalty ($1/number of grids$) will be applied for each move. The maze environments are configured in different sizes: 3x3, 5x5, 15x15.

Table 1 records the behavioral statistics after the highway graph RL agent solved the puzzle in each maze. The policy, derived from the highway graph, is considered to have converged only if newly collected experiences do not enhance the highway graph for better action generation. Although the time used varies, the optimal path to the desired location was found in all environments by the agent after a single iteration of value learning, and the training converged to the optimal policy within a minute. The state space of this kind of environment is relatively tiny (i.e. less than hundreds of states), so the highway can be constructed with no more than half a million frames and fulfill value learning entirely on the GPU in parallel. The value learning

Table 2: The HG behavior on Toy Text. We evaluate the convergence of value updating on the highway graph on three Toy Text environments to show the total reward and average discount return after convergence, frames and time taken until convergence, as well as the minimum number of learning iterations required for the convergence.

|  | Total Reward | Discounted avg. return | Frames used (M) | Wall clock time (min) | Converged learning iteration(s) |
|---|---|---|---|---|---|
| Taxi | 7.54±6.33 | 5.12±5.99 | 0.11±0.01 | 0.24±0.01 | 1 |
| CliffWalking | -13.00±0.01 | -12.88±0.01 | 0.10±0.01 | 0.30±0.02 | 1 |

on the whole graph, instead of sampling part of the graph, makes sure the optimal value will be propagated accurately to all states after each value learning iteration.

Next, to extend Table 1 with more training details, we compare the expected return over frames and clock wall time of our method and other baselines, shown in Figure 8. The highway graph RL method is 20 to more than 1000 times faster than other baselines to solve the Simple Maze 3x3, achieves the best expected return on Simple Maze 5x5, and is the only method to solve the Simple Maze 15x15 within one million frames. Both the model-free DQN, model-based MuZero, Stochastic MuZero, and Gumbel MuZero demonstrate instability and relatively slow value learning. Gumbel MuZero has been shown to outperform other MCTS methods in terms of planning efficiency, as it requires fewer simulations to achieve superior outcomes. They are unable to find a path to the desired location in Simple Maze 5x5 and 15x15 with such a few frames. This is because the sampled transitions may not always be in the optimal path when updating state values, preventing the optimal values from being propagated. This can be correct with much more sampling, but if the amount of sampled transitions is small, it is hard to update by the optimal value for states. Model-free policy gradient methods PPO and A3C outperformed MuZero and its variants in Simple Maze 3x3 and 5x5 with much fewer frames and time. However, PPO and A3C did not find the optimal path of Simple Maze 5x5 and 15x15 within one million frames. The reason is that the trajectories, used to update the state values, to the desired location sampled by PPO and A3C are not enough, and there is still improving space to obtain more precise state values. See Section 4.3 for more detailed analyzes.

Consequently, our HG can update state values with values from the optimal future states to avoid inaccurate and redundant value updates for every learning iteration.

### 4.1.2 Toy Text

Toy Text is more complex than Simple Maze since there will be many more paths to the desired location (Towers et al., 2023). We choose the deterministic CliffWalking and Taxi environments from Toy Text to evaluate the RL methods.

HG acts similarly when it is on Simple Maze, shown in Table 2. HG agent converged to the optimal policy and the optimal solution was found after one learning iteration within a minute.

The execution time and expected return of all methods are shown in Figure 9. HG achieved the best expected returns for all environments, and the growth rate of time over frames is the smallest among all methods, according to the third column in Figure 9. For CliffWalking results in Figure 9 (a), the optimized root action selection for MCTS helps Gumbel MuZero firstly converted to a locally optimal policy which is worse than the optimal policy due to the lack of sampling and training. DQN and PPO gradually increased the ability to get higher scores until the global optimal was reached. A3C and Stochastic MuZero did not effectively find a solution within one million frames of training. CliffWalking is different from Simple Maze since most grids have paths to the desired location. Thus, most sampled transitions of DQN can propagate optimal values to the starting state. However, randomly sampled trajectories will fade and slow down the optimal value propagation of A3C for each state. For Taxi, all baselines were unable to find the solution within one million frames, see Figure 9 (b). MuZero, Gumbel MuZero, A3C, and DQN converged to a local optimal policy. PPO and Stochastic MuZero require more training to develop better behavior.

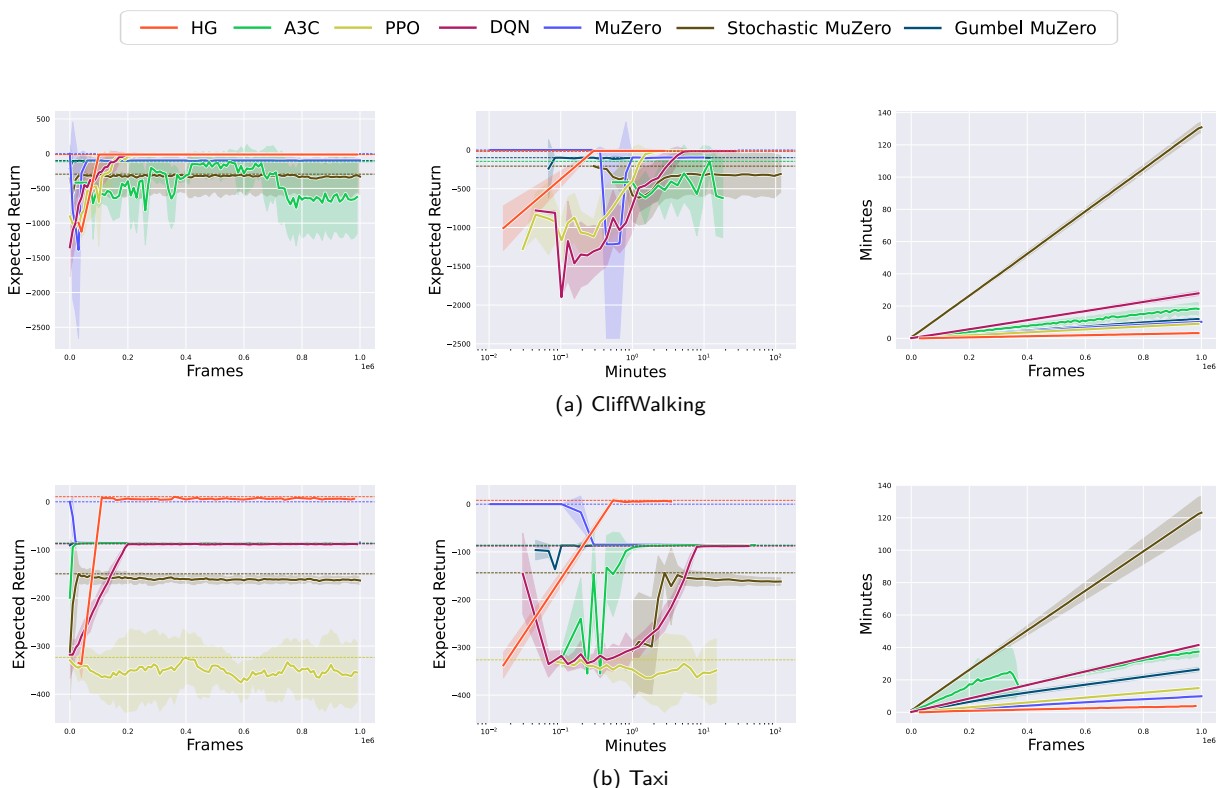

Figure 9: Episodic reward and efficiency comparison on Toy Text environments. The lines and shaded areas are for the average values and the corresponding range (±) of standard deviations, and dashed horizontal lines show the asymptotic performance. Each run of every method was recorded from three perspectives simultaneously: frames and minutes versus expected return to demonstrate the performance over frames and time (in the first two columns), and frames versus minutes to illustrate the relative training efficiency (in the last column).

Thus, these environments are again examples to show our HG can update state values without inaccuracy and redundancy.

### 4.1.3  Google Research Football

Next, we test our algorithm on the more difficult and computationally demanding Google Research Football (GRF) (Kurach et al., 2020) environment to show the efficiency of value updating. Agents are evaluated on three customized environments of 3 vs 2, 5 vs 5, and 11 vs 11 with increasing difficulty. The 3v2 is the attacking scenario with three defenders and two offenders including one goalkeeper for both teams. The tasks of 5v5/11v11 are confrontation scenarios with 5/11 players on each side. We use the single RL agent to control all the players in the team on the left-hand side according to the player's specific state at each time step, and the state is the SMM (Super Mini Map) feature (72 x 96 x 4) direct output from the GRF environment. Random seeds are used for environments to meet the deterministic assumption. There are two different rewards: goal reward and distance reward. If a team scores a goal, the simulator returns a +1.0 reward. The distance reward is determined by the minimum distance to the opponent's goal any team player ever made in this episode (the closer, the higher, with max +1.0).

The expected return and execution time of different RL agents are shown in Figure 10. The HG achieved the best expected return and scored for all scenarios within one million frames. Both IMPALA and R2D2 cannot score goals in all scenarios because the input of these environments is more complex and requires more computation to update the value of states. IMPALA made more progress on all scenarios compared with R2D2. This is partially because, in a large state space, the states on the path to the goal have a

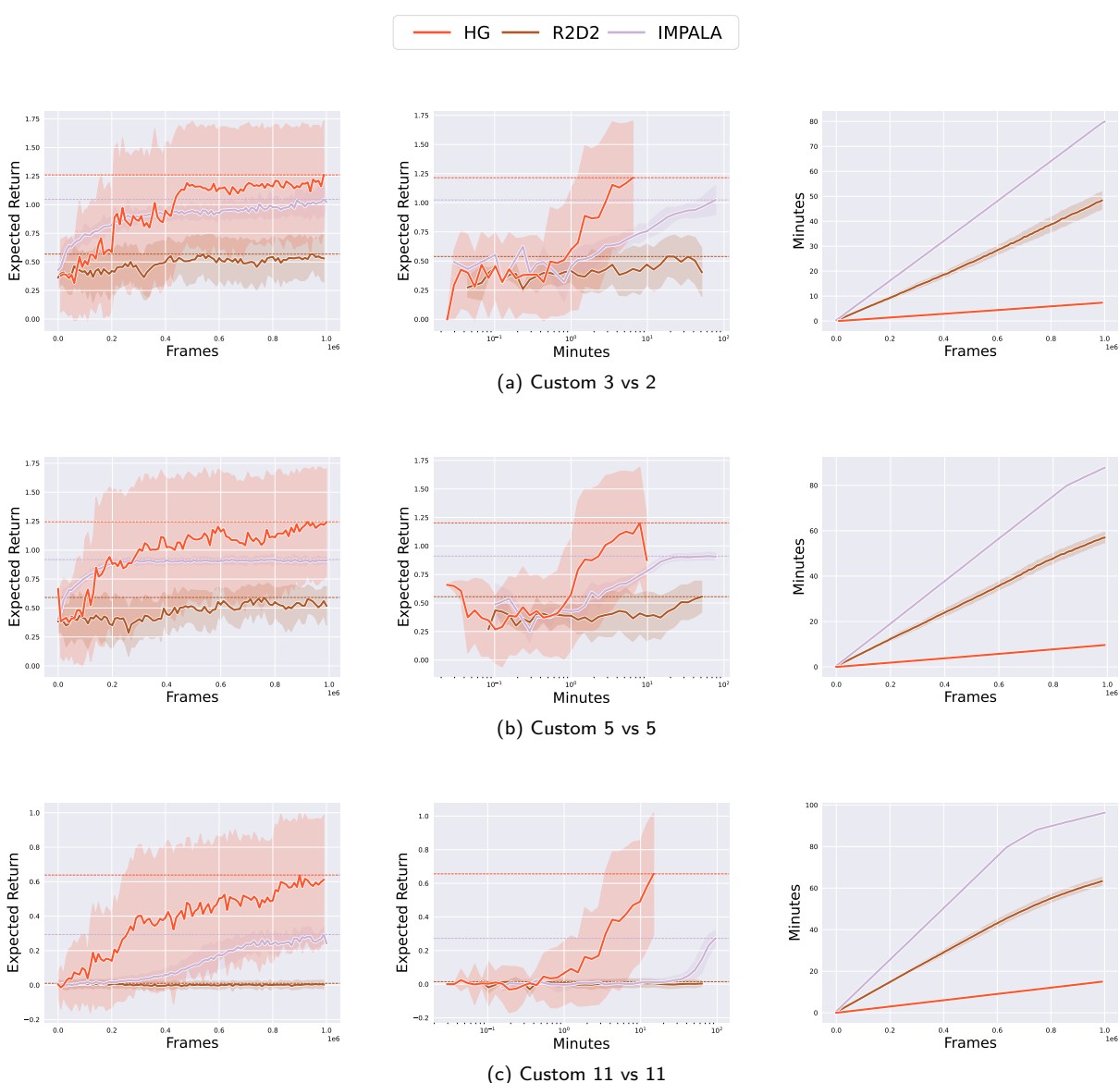

Figure 10: Expected return and efficiency comparison on 3v2, 5v5, 11v11 scenarios. The lines and shaded areas are for the average values and the corresponding range ($\pm$) of standard deviations, and dashed horizontal lines show the asymptotic performance. Each run of every method was recorded from three perspectives simultaneously: frames and minutes versus expected return to demonstrate the performance over frames and time (in the first two columns), and frames versus minutes to illustrate the relative training efficiency (in the last column).

higher chance of being chosen to update value by a policy gradient method than the Q-learning method. With the number of possible states becoming larger from 3 vs 2 to 11 vs 11, the learning efficiency of R2D2 correspondingly decreased.

From the result on GRF environments, we can see the efficiency advantage of our highway graph RL agent. See Section 4.3 for more speed advantage of value updating.

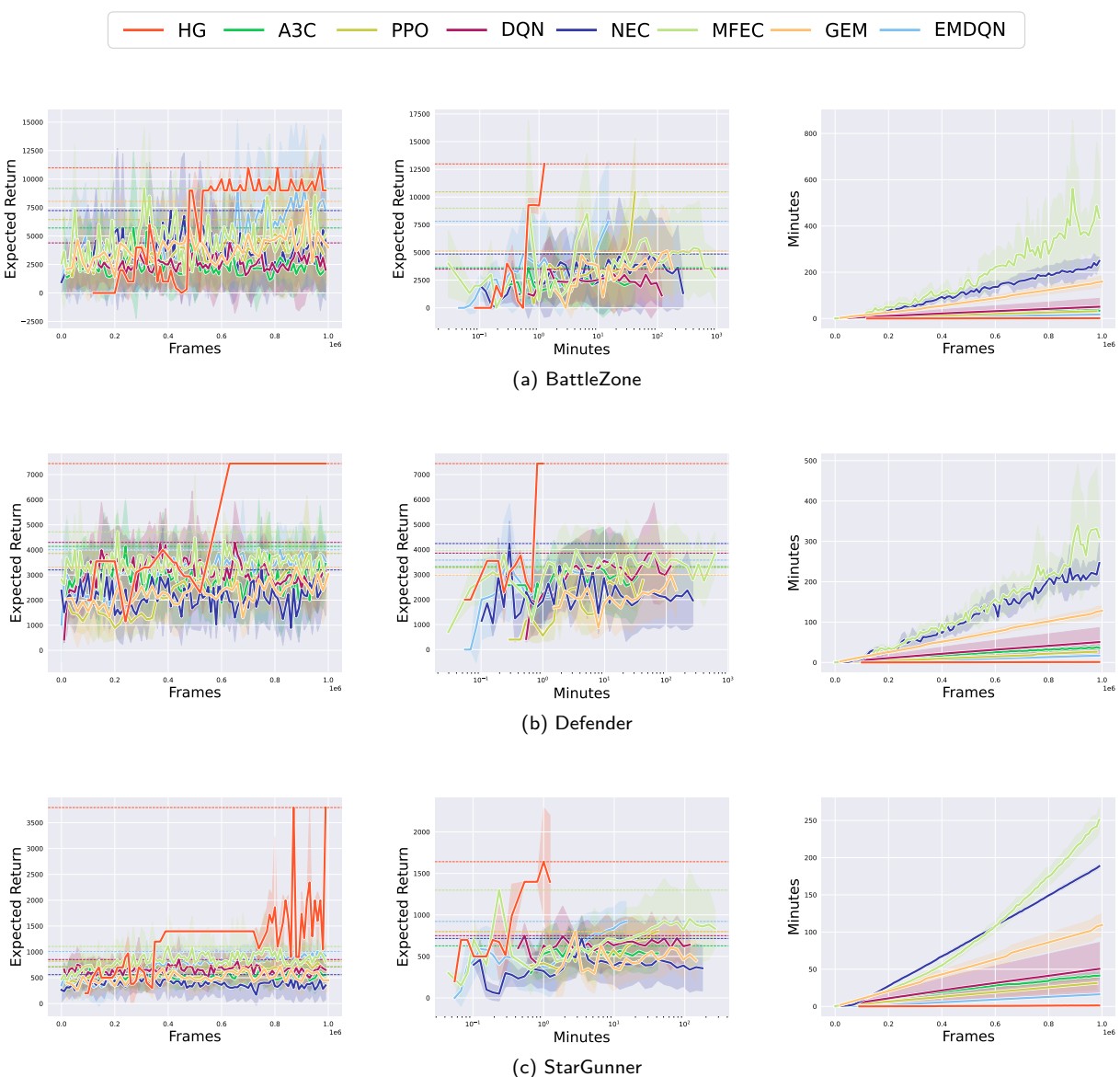

Figure 11: Expected return and efficiency comparison on Atari games. The lines and shaded areas are for the average values and the corresponding range (±) of standard deviations, and dashed horizontal lines show the asymptotic performance. Each run of every method was recorded from three perspectives simultaneously: frames and minutes versus expected return to demonstrate the performance over frames and time (in the first two columns), and frames versus minutes to illustrate the relative training efficiency (in the last column).

### 4.1.4 Atari learning environment

The Atari games using the Atari Learning Environment (Bellemare et al., 2013) are also used to evaluate our proposed method. We randomly select 3 games with mainstream genera of Atari to train the different RL agents for one million frames. An environment seed is also used during initialization for all games to guarantee the deterministic property.

The state of Atari games may not be directly accessible, instead we got an observation from the environment. To solve this, we use a recurrent random linear projector on all the past observations of the episode, to obtain a more compact state representation for the highway graph. This projection does not affect the deterministic property of the environments. More details of the random linear projector are in the Appendix.

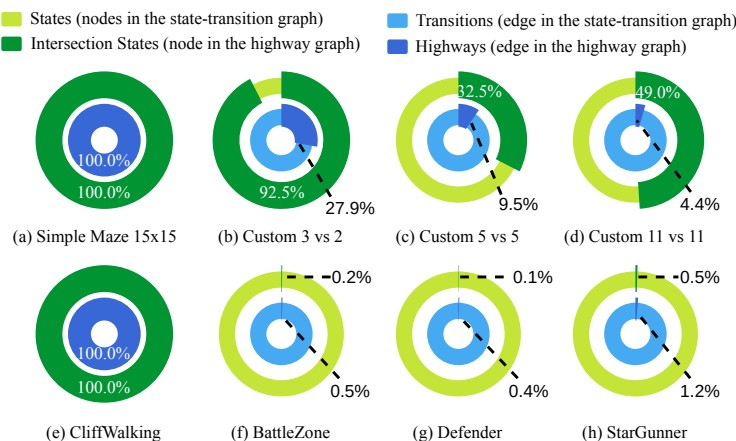

Figure 12: Percentage of nodes and edges in the highway graph compared with its corresponding state-transition graph. The conversion into highway graphs dramatically reduces the size of both nodes and edges.

Next, we compare the total scores of different agents on different games, shown in Figure 11. We can see the expected return curve of HG is above all the baselines for every environment. It shows the value of states of a trained HG agent will be more accurate than other methods. Another advantage of HG is drawn on the efficiency. The growth rate of time over frames of HG is the smallest. Additionally, HG only used about 1/10 to less than 1/150 period of time compared to other baselines to get converged and finish the training.

Many episodic-control-based methods including EMDQN and MFEC show a high sample efficiency by achieving relatively high scores, but the training efficiency of these methods is low.

Consequently, the highway of HG improves the value updating efficiency and accuracy simultaneously on Atari games.

## 4.2 Structural advantage of highway graph

To investigate the advantage of our highway graph compared with the original empirical state-transition graph, we compare the size of nodes and edges for different environments, see Figure 12. The size of the nodes and edges of the highway graph is smaller than the original empirical state-transition graph. For example, the empirical state-transition graph of StarGunner has 234,521 nodes and 236,266 edges, whereas its highway graph only has 1,084 nodes (0.5%) and 2,829 edges (1.2%). According to Remark 1, $z = 0.005 \ll 1$ means the computation required by the value updating will be theoretically reduced to $\frac{1}{2.5 \times 10^5}$. Many environments, including all evaluated Atari games, have very few intersection states (nodes), which makes the highway graph small enough to be loaded entirely into GPU to update all known state-action values in parallel, and hence achieve a promising acceleration during training.

Simple tasks, such as Simple Maze and Toy Text environments, have a densely connected state-transition graph with a small state space of the empirical state-transition graph, whereas tough tasks observe a nearly tree-like state-transition graph structure, where states on different branches are not connected, with a large state space. This trend becomes more obvious as the complexity of the environment increases, since from a state, in a complex environment, it is not easy to move to a previously visited state. After converting to highways, many paths can be shrunk as highways to reduce the number of states and edges. Thus, the use of the highway graph to model and update the value can be less computationally extensive. A comparison of a highway graph and its original empirical state-transition graph of a trained agent for the Custom 5 vs 5 is shown in Figure 13. More comparison for the Atari game can be found in the Appendix.

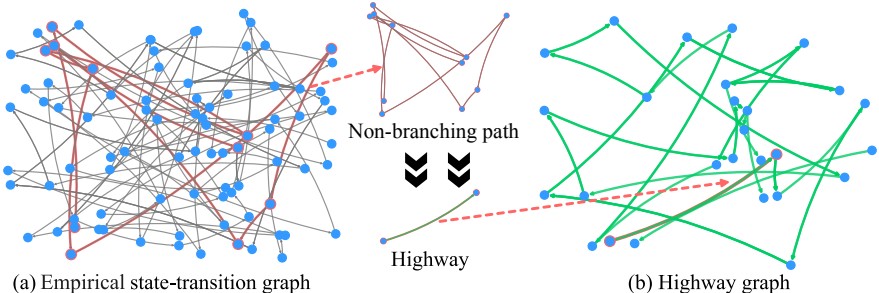

(a) Empirical state-transition graph          (b) Highway graph

Figure 13: The empirical state-transition graph (a) and its corresponding highway graph (b) of Custom 5 vs 5. The states in both (a) and (b) are blue cycles, highways in (b) are green arrows, and transitions are in grey. One of the highways and its corresponding path are highlighted in red in (b) and (a) respectively. The nodes and edges in the highway graph are considerably reduced.

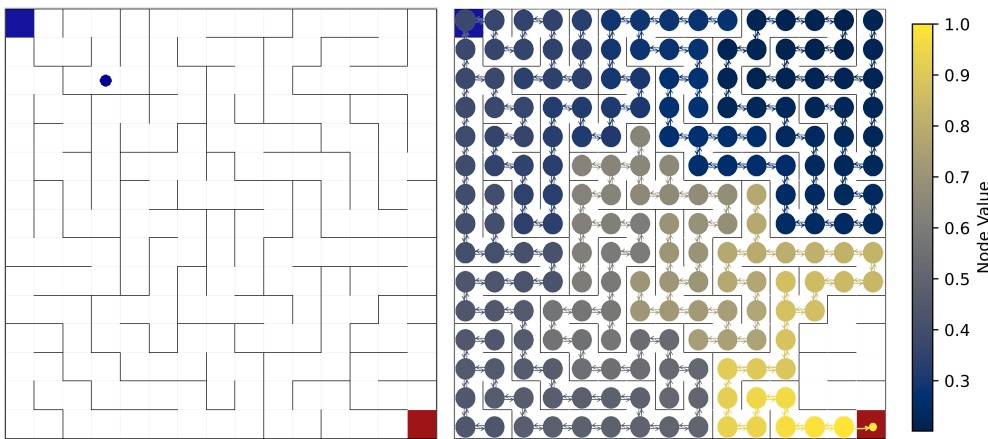

Figure 14: The 15x15 maze environment (left) and its state values on the corresponding highway graph (right, self-loops are not shown).

## 4.3 Value updating advantage of highway graph

**Value updating completeness**   This experiment investigates the distance between the values output by trained RL agents and the ground truth values of each state on Simple Maze 15x15 (225 states in total). Completeness indicates the percentage of state values that are identical to the ground truth. The distances and completeness of value updating of different methods are shown in Table 3. From the results, our method performs optimal value updating of this task, and all states' values are updated ideally. The visualization of the learned HG model for the Simple Maze with a size of 15x15 is shown in Figure 14. The comparison methods cannot completely update the values of all states and many states have incorrect values.

In the case of Simple Maze 15x15, the DQN cannot successfully find a path to the target location. We find this is caused by the design of Q-learning. The effective training of Q-learning-based algorithms relies on a considerable amount of high-rewarded samples in the replay buffer (the ratio of the high-rewarded samples is task-dependent). The training of the state-action value function of this kind of method is influenced by the sampled transition and its built-in target state-action value function. Using the high-rewarded samples, the state-action value function will be more close to the real situation. Later, the target state-action value function can be updated more reliably, and the agent training enters a virtuous cycle. However, if the high-rewarded sample is not enough, the target state-action value function will take more effect. The sampling in the Simple Maze environment is not necessary to always find high-rewarded samples, and more exploration may find samples with lower rewards due to moving penalties. That leads to an unusable target state-action value function at an early stage so that the entire Q-learning will fail.

Table 3: Value updating completeness. It shows the distance to the ground truth of the learned value after one million frames of training. Completeness indicates the percentage of state value that is identical to the ground truth.

| Method | Min Distance | Max Distance | Avg. Distance | Completeness (%) |
|---|---|---|---|---|
| PPO | 0.24 | 0.99 | 0.53 | 1.88% |
| DQN | 0.13 | 0.92 | 0.41 | 6.57% |
| HG (Ours) | 0.00 | 0.00 | 0.00 | 100.00% |

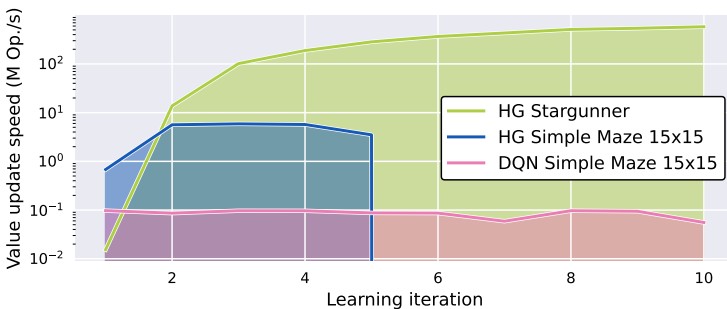

Figure 15: The million step/operation (Op.) per second of different value update iterations. The value update speed is the value propagation steps among the original empirical state-transition graph per training iteration. In each iteration, the highway graph is converted to an empirical transition graph to calculate the value update speed.

**Value updating speed advantage**  We further show the value updating speed (the value propagation steps among the original empirical state-transition graph per training iteration). We use the Simple Maze with the size of 15 x 15 (225 states in total) and StarGunner to show the value updating speed. The results are in Figure 15. The speed of value updating on the highway graph is converted to value updating steps in the empirical state-transition graph, by counting the value updating operation for each transition. The results show the value updating speed of DQN is the slowest which is due to the N-step TD learning algorithm, where only parts of the states are sampled for the value learning. On the contrary, our HG always updates the entire graph, and HG reaches the $1159 \pm 31$ million Op./s value updating speed during the 10th value update in the StarGunner. HG's value update speed gives evidence for faster convergence on previously evaluated environments.

## 4.4   Performance of re-parameterized neural network-based agent

**Graph re-parameterization**  Atari games require very large state-transition graphs to store the environmental information, and graph re-parameterization can help to reduce storage costs. To re-parameterize the information in the highway graph, we use a very simple 2-layered MLP with 512 units in each layer to store the state-action value of the highway graph. The re-parameterized neural network-based agent is HG-Q. The relative performance of the expected return of the neural network-based agent HG-Q compared with the HG is shown in Figure 16.

From the result, we can see the neural network-based agent achieved an on-par performance compared with the agent built from the highway graph. Additionally, on Atari games, the neural network-based agent outperformed the highway graph by about 5% to 20%, which shows the generalization ability of the neural networks. These experiments indicate that the highway graph can be a jump-starter for other RL agents during the early stages of training.

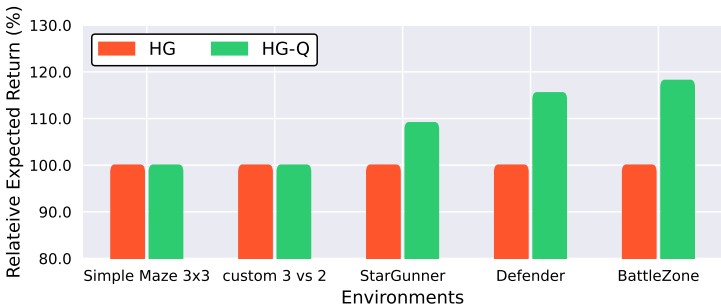

Figure 16: The relative expected return of HG-Q compared with HG.

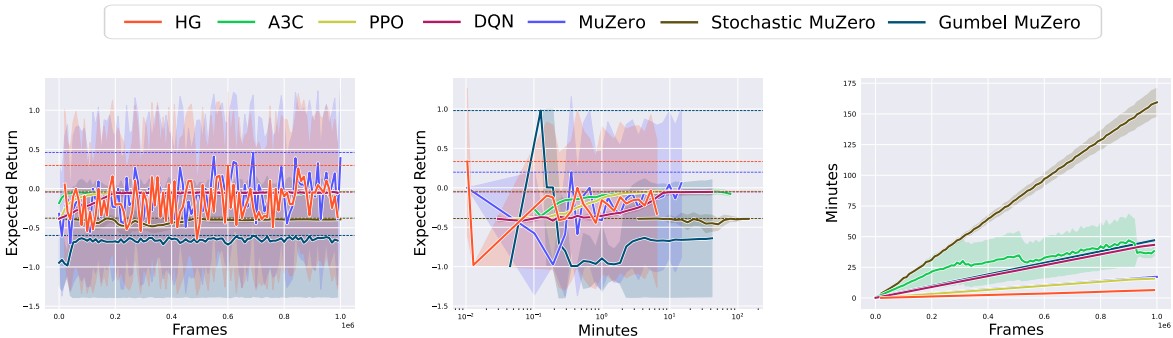

Figure 17: Highway graph on stochastic environments Blackjack.

## 5 Limitations

One of the important assumptions when designing the highway graph is that the state can be restored by observation. Thus, the highway graph does not suit partially observable and stochastic environments. (1) In a stochastic environment, from a specific state, the same action does not always lead to the same future state. In this case, the value updating of the highway graph will not be correct since the value will not always propagate in the same way. (2) In a partially observable environment, the state in the state-transition graph will not always be reliable, one state in the graph might link to two different real states, i.e. state conflict. When state conflict exists, value updating of the highway graph will be incorrectly caused by the same issue in the stochastic environment.

Experimental results on Blackjack support this idea. The goal for players of Blackjack in Figure 17 is to obtain 2 cards that sum to closer to 21 (without going over 21) than the dealer's cards. The card the player gets every time is randomly selected, which makes the environment stochastic. We evaluate the expected return of the highway graph of stochastic environment Blackjack from Toy text and the expected return will be unpredictable, resulting in a failed highway graph, see Figure 17. The found result is mainly caused by the state conflict, which means two different states will have the same representation and use the same node in the highway graph. The different nodes in the highway graph should lead to different actions, but the state conflict prevents the agent from doing so.

## 6 Conclusions

The high cost of training RL agents has constrained their research and practical applications. To address this challenge, we explored a novel approach to significantly reduce this cost by applying the concept of highways, inspired by real-world transportation systems, to the value-updating process of RL algorithms. We introduced the highway graph structure, derived from the empirical state-transition graph, which substantially reduces the state space. This structure can be learned from sampled transitions and seamlessly

integrated into the value iteration procedure, enabling an efficient value-updating process. Furthermore, we provided theoretical proof of convergence to the optimal state value function when using the highway graph for value learning.

In our experiments, we evaluated the highway graph RL method across four categories of learning tasks, benchmarking it against a range of other RL methods. The results demonstrate that our method drastically reduces training time, achieving speedups ranging from 10-fold to over 150-fold compared to baselines, while maintaining equal or superior expected returns. Additionally, we re-parameterized the highway graph into a simple neural network-based agent, which achieves comparable or improved performance with minimal storage requirements.

To enhance the versatility of the highway graph RL method, future work could extend the approach to partially observable environments with stochastic characteristics. Moreover, the highway graph's aggregated and efficient value updating process could potentially be integrated into various RL algorithms during value learning, including value-based algorithms for multi-agent RL (Albrecht et al., 2024). By doing so, the highway graph has the potential to accelerate the value learning process of a broader range of RL algorithms.

**Acknowledgments**

This work was supported by the Basic Research Foundation for Youths of Yunnan Province (202301AU070147).

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

## A Details of baselines

For Simple Maze, Toy text, and Google Research Football, we adopt a discount factor of 0.99, and for Atari games, we adopt a discount factor of $1 - 1e6$ since very long trajectories in Atari games are not suitable for recurrent random projectors.

The official or widely recognized implementations, based on the original papers, for MuZero, Gumbel MuZero, and Stochastic MuZero are adopted, and the grid search was conducted on the most critical hyperparameters for them to ensure we obtained the best results. We use the default hyperparameters for those from the official code including NEC, MFEC, EMDQN, and GEM, which were tested to be the optimal ones according to its original paper. Other methods are implemented by the RLlib, and we adopt the best hyperparameters by fine-tuning and referencing the previous evaluation reported in the RLlib.

The network structure of different baselines depends on the environment observation spaces. If the observation is images (Atari games), a CNN encoder is attached to the front of the network, otherwise, a fully connected encoder is attached instead. For R2D2 and A3C, an LSTM encoder is used. More detailed network architectures and hyperparameters of different baselines are:

- DQN: double DQN is used with dueling enabled. $N$-step of Q-learning is 1. A Huber loss is computed for TD error.

- A3C: the coefficient for the value function term and the entropy regularizer term in the loss function are 0.5 and 0.01 respectively. The grad clip is set to 40.

- PPO: initial coefficient and target value for KL divergence are 0.5 and 0.01 respectively. The coefficient of the value function loss is 1.0.

- R2D2: the LSTM cell size is 64 and the max sequence length is 20.

- IMPALA: the coefficient for the value function term and the entropy regularizer term in the loss function are 0.5 and 0.01 respectively.

- MuZero: the implementation uses four 16-layer MLPs to learn the transition, reward function, state value function, and policy to select the action.

- Gumbel MuZero: the implementation uses MLPs to learn the representation, dynamic, and prediction functions to serve the planning and action selection processes.

- Stochastic MuZero: the implementation utilizes a 4-layer MLP as an encoder with a hidden size of 128 for the observation, and the rest of the parts remain as default.

- NEC: the network structure consists of a 3-layer CNN followed by a linear layer. The size of the DND dictionary is 500000, $\alpha$ for updating stored values is 0.1, and $\delta$ for thresholding the closeness of neighbors is 0.001.

- EMDQN: there are 3 convolution layers and 2 linear layers in the network.

- GEM: the network is constructed using 3 convolution layers and 1 linear layer.

## B Random projection

Following the episodic control methods (Blundell et al., 2016), we use a mapping function $F_{\mathcal{S}}$, such as RNN, or CNN+RNN with randomly initialized fixed parameters, to obtain the state representation of MDPs, i.e. random projection. Random projection transforms the input to another lower dimensional space with the linear properties preserved. Although learnable functions show promising results (Mnih et al., 2015; 2016), the state representation is always changing during the entire training process, which increases the difficulties of convergence. Instead, random projection without a learning process will produce fixed state

representations during the entire training, but its ability to represent complicated relationships between states is less powerful.

For Atari games, we adopt a simple static linear RNN to reduce the state uncertainty by using all the historical observations, and each state $s_t$ in the trajectory $l$ can be calculated by Equation 16.

$$[s_t, \mathbf{h}_t] = \mathbf{w}_{random} * (o_t, \mathbf{h}_{t-1}) + \mathbf{b}_{random}(o_t \in l) \tag{16}$$

where $\mathbf{h}_{t-1}$ is the hidden vector of RNN. First hidden vector $\mathbf{h}_{-1}$ is randomly initialized. $\mathbf{w}_{random}$ and $\mathbf{b}_{random}$ is initialized with xavier initializer (Glorot & Bengio, 2010), and will not update afterwards.

## C  Proof of Lemma 1

*Proof.* According to Equation 1, the value distance of any two different state value function estimates $\hat{V}^1$ and $\hat{V}^2$ for a state $s_i \in \mathcal{S}_{inter}$ can be described by

$$|\hat{V}^1(s_i) - \hat{V}^2(s_i)| = \left| \max_a \left[ \mathcal{R}(s_i, a) + \gamma \sum_{s_i'} \mathcal{T}(s_i', a, s_i) \hat{V}^1(s_i') \right] - \max_a \left[ \mathcal{R}(s_i, a) + \gamma \sum_{s_i'} \mathcal{T}(s_i', a, s_i) \hat{V}^2(s_i') \right] \right|.$$

For any two different points $w, v \in \mathcal{V}$ ($\mathcal{V} \in \mathbb{R}^{|\mathcal{S}_{inter}|}$), we have two state value vectors

$$w = \left[ \hat{V}^1(s_1), \hat{V}^1(s_2), \cdots, \hat{V}^1(s_i), \cdots \right]^\intercal,$$
$$v = \left[ \hat{V}^2(s_1), \hat{V}^2(s_2), \cdots, \hat{V}^2(s_i), \cdots \right]^\intercal,$$

where $s_i, s_j \in \mathcal{S}_{inter}$. The difference between $w$ and $v$ can be described by the metric $d$ as

$$
\begin{aligned}
d(w, v) &= \|\hat{V}^1(\mathcal{S}_{inter}) - \hat{V}^2(\mathcal{S}_{inter})\|_\infty, \\
&= \max_i |\hat{V}^1(s_i) - \hat{V}^2(s_i)| \\
&= \max_i \left| \max_a \left[ \mathcal{R}(s_i, a) + \gamma \sum_{s_i'} \mathcal{T}(s_i', a, s_i) \hat{V}^1(s_i') \right] - \max_a \left[ \mathcal{R}(s_i, a) + \gamma \sum_{s_i'} \mathcal{T}(s_i', a, s_i) \hat{V}^2(s_i') \right] \right|.
\end{aligned}
$$

Thus, we denote $(\mathcal{V}, d)$ the metric space from highway MDP.

Next, we prove all Cauchy sequences of $\mathcal{V}$ have a limit. Let $(v^{(m)})$ and $(v^{(r)})$ be any two different Cauchy sequences of $\mathcal{V}$, i.e.

$$(v^{(m)}) = (v_1^{(m)}, v_2^{(m)}, \cdots, v_{|\mathcal{S}_{inter}|}^{(m)}),$$
$$(v^{(r)}) = (v_1^{(r)}, v_2^{(r)}, \cdots, v_{|\mathcal{S}_{inter}|}^{(r)}).$$

According to the definition of the Cauchy sequence (Khamsi & Kirk, 2011), for arbitrary $\epsilon > 0$, there exist $N$ when $m, r > N$ that

$$d\left(v^{(m)}, v^{(r)}\right) = \max_i \left| v_i^{(m)} - v_i^{(r)} \right| < \epsilon.$$

For any $i$, $\left| v_i^{(m)} - v_i^{(r)} \right| < \epsilon$, so $(v_i^{(m)})$ and $(v_i^{(r)})$ are Cauchy sequences of $\mathbb{R}$.

Since $\mathbb{R}$ is a complete metric space (Khamsi & Kirk, 2011), for arbitrary $i$, $v_i^{(r)}$ has a limit. Thus, we denote $\lim_{r \to \infty} v_i^{(r)} = v_i^*$, and we have $(v^*) = (v_1^*, \cdots, v_{|\mathcal{S}_{inter}|}^*)$, which means every element of $(v_i^{(r)})$ converged to a fixed point.

According to the definition of the Cauchy sequence, we have

$$d\left(v^{(m)}, v^*\right) \quad = \quad \max_i \left| v_i^{(m)} - v_i^* \right| < \epsilon.$$

The above relation shows that any Cauchy sequence of $\mathcal{V}$ converges to a fixed point.

Hence, $(\mathcal{V}, d)$ is a complete metric space. □

## D    Proof of Lemma 2

*Proof.* For any two different state value function estimates $\hat{V}^1$ and $\hat{V}^2$, we have two state value vectors

$$w = \left[ \hat{V}^1(s_1), \hat{V}^1(s_2), \cdots, \hat{V}^1(s_i), \cdots \right]^\mathsf{T},$$
$$v = \left[ \hat{V}^2(s_1), \hat{V}^2(s_2), \cdots, \hat{V}^2(s_i), \cdots \right]^\mathsf{T},$$

where $s_i \in \mathcal{S}_{inter}$, and we study the metric $d$ for any $s_i \in \mathcal{S}_{inter}$

$$
\begin{aligned}
d\left(G(\hat{V}^1(s_i)), G(\hat{V}^2(s_i))\right) &= \left\| \Phi_{s_j \in \Gamma_i^1}\left(\mathcal{R}(s_i, a) + \gamma\hat{V}^1(s_j)\right) - \Phi_{s_j \in \Gamma_i^1}\left(\mathcal{R}(s_i, a) + \gamma\hat{V}^2(s_j)\right) \right\|_\infty \\
&\leq \Phi_{s_j \in \Gamma_i^1}\left\| \left(\mathcal{R}(s_i, a) + \gamma\hat{V}^1(s_j)\right) - \left(\mathcal{R}(s_i, a) + \gamma\hat{V}^2(s_j)\right) \right\|_\infty \\
&= \gamma\Phi_{s_j \in \Gamma_i^1}\left\| \hat{V}^1(s_j) - \hat{V}^2(s_j) \right\|_\infty \\
&\leq \gamma\left\| w - v \right\|_\infty.
\end{aligned}
$$

The metric for all intersection states $\mathcal{S}_{inter}$ will be

$$d\left(G(w), G(v)\right) \leq \gamma \left\| w - v \right\|_\infty.$$

Thus, the contraction property of the graph Bellman operator $G$ holds that

$$d\left(G(w), G(v)\right) \leq \gamma d\left(w - v\right),$$

and $G$ is a max-norm $\gamma$-contraction mapping. □

## E    Proof of Proposition 1

*Proof.* The following proof will firstly give the convergence property for value updating on the highway graph, and prove the uniqueness of the convergence.

**Property of convergence**    Given the complete metric space $(\mathcal{V}, d)$, for arbitrary $w, v, x \in \mathcal{V}$, the following properties (symmetry and triangle inequality) (Khamsi & Kirk, 2011) hold

$$
\begin{aligned}
d\left(w, v\right) &= d\left(v, w\right), \\
d\left(w, x\right) &\leq d\left(w, v\right) + d\left(v, x\right).
\end{aligned}
$$

Lemma 2 has proved the graph Bellman operator $G$ a $\gamma$-contraction mapping for $\mathcal{V}$, i.e. $d\left(G(w), G(v)\right) \leq \gamma d\left(w, v\right)$, then for arbitrary $w, v \in \mathcal{V}$, we have

$$
\begin{aligned}
d\left(w, v\right) &\leq d\left(w, G(w)\right) + d\left(G(w), v\right) \\
&\leq d\left(w, G(w)\right) + d\left(G(w), G(v)\right) + d\left(G(v), v\right) \\
&\leq d\left(G(w), w\right) + \gamma d\left(w, v\right) + d\left(G(v), v\right).
\end{aligned}
$$

Moving the term $\gamma d(w, v)$ to the left and dividing $1 - \gamma$ for both sides, we have

$$d(w, v) \leq \frac{d(G(w), w) + d(G(v), v)}{1 - \gamma}$$

Next, for any $w \in \mathcal{V}$ we define a sequence $(G^{(n)})$

$$\left( G^{(2)}(w) = G(G(w)), G^{(3)}(w) = G(G^{(2)}(w)), \cdots, G^{(n)}(w) = G(G^{(n-1)}(w)) \right).$$

Then, we study any two elements of the defined sequence $G^{(i)}(w), G^{(j)}(w) \in (G^{(n)})$ that

$$
\begin{aligned}
d\left( G^{(i)}(w), G^{(j)}(w) \right) &\leq \frac{d\left( G^{(i+1)}(w), G^{(i)}(w) \right) + d\left( G^{(j+1)}(w), G^{(j)}(w) \right)}{1 - \gamma} \\
&\leq \frac{\gamma^i d(G(w), w) + \gamma^j d(G(w), w)}{1 - \gamma} \\
&= \frac{\gamma^i + \gamma^j}{1 - \gamma} d(G(w), w).
\end{aligned}
$$

Since $\gamma \in [0, 1)$, $i, j \to \infty$ leads to $\frac{\gamma^i + \gamma^j}{1 - \gamma} d(G(w), w) \to 0$. So $(G^{(n)})$ is a Cauchy sequence of $\mathcal{V}$. Given $(\mathcal{V}, d)$ is a complete metric space proved by Lemma 1, where all Cauchy sequences will converge within $\mathcal{V}$, the property of convergence of $(G^{(n)})$ hold consequently.

**Uniqueness of convergence** Assume the value updating of the state value function converged to some point $w^* \in \mathcal{V}$, which satisfies $w^* = \lim_{n \to \infty} G^{(n)}(w)$. According to the graph Bellman operator $G$, we need to find that

$$
\begin{aligned}
G(w^*) &= G\left( \lim_{n \to \infty} G^{(n)}(w) \right) \\
&= \lim_{n \to \infty} G\left( G^{(n)}(w) \right) \\
&= \lim_{n \to \infty} G^{(n+1)}(w) \\
&= w^*.
\end{aligned}
$$

We can use proof by contradiction. Assume there are two different points $w^*, v^* \in \mathcal{V}(w^* \neq v^*)$ which satisfy $G(w^*) = w^*$ and $G(v^*) = v^*$, then we have

$$
\begin{aligned}
0 &< d(w^*, v^*) \\
&= d(G(w^*), G(v^*)) \\
&\leq \gamma d(w^*, v^*) \\
&< d(w^*, v^*).
\end{aligned}
$$

We find it contradictory that $d(w^*, v^*) < d(w^*, v^*)$, and therefore conclude that $G(w^*) = G(v^*) = w^*$.

Furthermore, we study that

$$
\begin{aligned}
d\left( G^{(n)}(w), w^* \right) &= d\left( G^{(n)}(w), G(w^*) \right) \\
&\leq \gamma d\left( G^{(n-1)}(w), w^* \right) \\
&\leq \gamma^n d(w, w^*),
\end{aligned}
\tag{17}
$$

which indicates the sequence $(G^{(n)})$ has a linear rate of convergence determined by $\gamma$.

From above, there is only one fixed point

$$w^* = \left[ V_{\mathcal{H}}^*(s_1), V_{\mathcal{H}}^*(s_2), \cdots, V_{\mathcal{H}}^*(s_i), \cdots \right]^\top,$$

that the value updating will converge to, where $V_{\mathcal{H}}^*$ is the corresponding optimal state value function for the fixed point $w^*$ of convergence. This shows that the state value function $V_{\mathcal{H}}$ during value updating on the highway graph will converge to the optimal state value function $V_{\mathcal{H}}^*$.

$\square$

## F More discussions of Remark 1

The VI process aims to pass the value information among all states, which shares the value information of all states for every state. This process takes two steps: (1) interchanges value information pairwisely between any pair of states, which takes $|\mathcal{S}| \times |\mathcal{A}|$ number of computations, where $|\mathcal{S}|, |\mathcal{A}|$ is the number of states and actions; (2) passes the value information for a state from its farthest state by pairwise exchange in (1). The VI process in our work exhibits two extreme scenarios, with other cases lying in between these two.

**Case 1**. The entire empirical state-transition graph consists of a single long path made of non-branching transitions with $|\mathcal{S}|$ states, and the highway graph is reduced to two intersection states connected by a single edge. Note that in the non-branching sequence of transitions, $|\mathcal{A}|$ will be 1. In addition, the possible farthest information passing starts from the last state to the initial state in this path, and the number of computations will be $|\mathcal{S}|$. Therefore, the time complexity of VI on the empirical state-transition graph will be $(|\mathcal{S}| \times 1) \times |\mathcal{S}|$. Although the reduced states form the highway graph, the VI process does not change, and the difference is the size of the states reduced. So, the time complexity of VI for the highway graph will be $|\mathcal{S}_{inter}| \times 1 \times |\mathcal{S}_{inter}|$, where $|\mathcal{S}_{inter}| = |\mathcal{S}| * z$. That means the VI process on the highway graph will be $\frac{1}{z^2}$ faster compared with the empirical state-transition graph.

**Case 2**. Despite the rarity of this occurrence in RL environments, all states in the empirical state-transition graph can densely connect all the other states via an action. In this case, step (1) of VI will pass all the value information among all states, and step (2) will not be required. Since states connect to all the other states, $|\mathcal{A}| = |\mathcal{S}|$. The states are all intersections, which means $|\mathcal{S}| = |\mathcal{S}_{inter}|$. So, the time complexity of VI on both the empirical state-transition graph and highway graph will be $|\mathcal{S}| \times |A| = |S| \times |S| = \frac{|\mathcal{S}_{inter}| \times |\mathcal{S}_{inter}|}{z^2}$, where $z = \frac{|\mathcal{S}_{inter}|}{|\mathcal{S}|} = 1$. The condition is still satisfied that the highway graph is $\frac{1}{z^2}$ times faster during VI.

Other cases lie in between Case 1 and Case 2. Consequently, the reduction of time complexity of adopting the highway graph is in consensus with Remark 1.

## G More experimental results

### G.1 Highway graph for Pong

A comparison of a highway graph and its original empirical state-transition graph using 0.1 million frames for the Atari game Pong is shown in Figure 18. The size of the highway graph is significantly smaller than that of the original empirical state-transition graph, and this will expedite the value learning process during agent training.

### G.2 Exploration capability

To assess the exploration capability of our HG method, on Atari games, we reduce the number of episodes for periodically updating the highway graph to one-fifth of its original value and increase the frame budgets by five times to 5 million frames. Through this setting, it will be more evident whether HG is more prone to getting stuck in suboptimal trajectories. The results are presented in Figure 19 (a) - (c).

According to the results, it can be observed that the expected return of HG in all three games consistently increases without getting stuck at a certain point and outperforms the performance compared to that with 1 million frames.

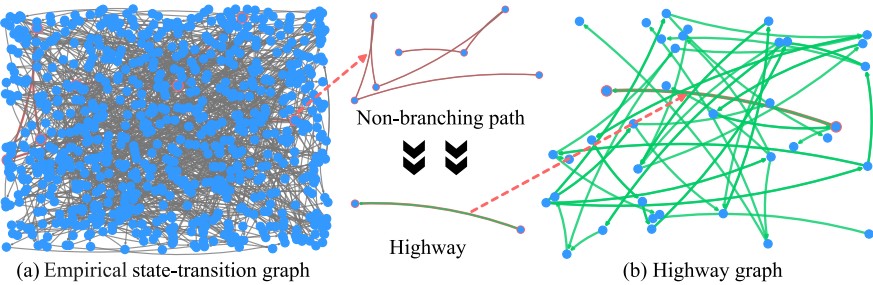

(a) Empirical state-transition graph          (b) Highway graph

Figure 18: The empirical state-transition graph (a) and its corresponding highway graph (b) of Pong. The states in both (a) and (b) are blue cycles, highways in (b) are green arrows, and transitions are in grey. One of the highways and its corresponding path are highlighted in red in (b) and (a) respectively. The nodes and edges in the highway graph are considerably reduced.

Next, we additionally assess HG on Pong. In this context, we reduce the number of episodes for periodically updating the highway graph to one-tenth and increase the frame budgets to 10 million frames. The results are presented in Figure 19 (d).

From the results, HG solved Pong within 15 minutes by using approximately 6 to 9 million frames.

Based on all the aforementioned results, it can be observed that HG has the ability to explore better trajectories even when the initial highway graph is extremely small.

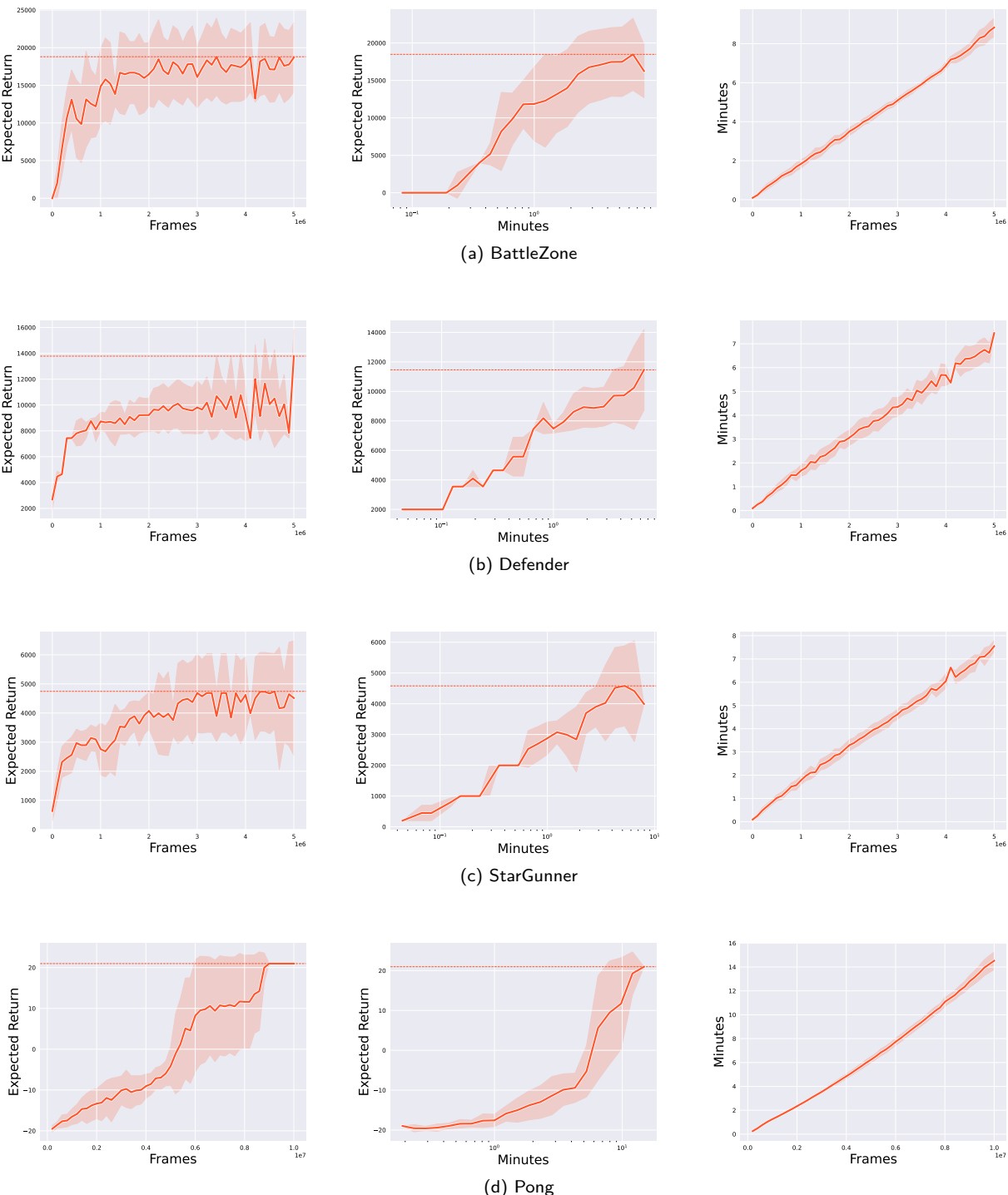

Figure 19: Expected return and efficiency of HG on Atari games. The lines and shaded areas are for the average values and the corresponding range ($\pm$) of standard deviations, and dashed horizontal lines show the asymptotic performance. Each run of every method was recorded from three perspectives simultaneously: frames and minutes versus expected return to demonstrate the performance over frames and time (in the first two columns), and frames versus minutes to illustrate the relative training efficiency (in the last column).

