# OpenReview forum: "Highway Graph to Accelerate Reinforcement Learning"
_TMLR — Accepted by TMLR_

### Review · Reviewer_C93Q · 2024-07-16

**Summary Of Contributions:**

The paper proposes a method to accelerate reinforcement learning in deterministic settings by compressing the state transition model into a "highway graph" so that signal can propagate many steps into the past, making credit assignment much easier. Experiments support the strength of the proposed method.

**Audience:**

Yes

**Broader Impact Concerns:**

None that I can see.

**Claims And Evidence:**

No

**Requested Changes:**

Please see Weaknesses. In particular the results seem strong, but I am having quite a hard time judging them without understanding the exact method in each section, for each figure, and besides the Simple Maze setting that does not seem to be currently in the text (and even then I have questions about the Simple Maze setting. Addressing both weaknesses would be critical for my acceptance. I would be happy also to engage in dialogue around both points as I need more information before making recommendations.

**Strengths And Weaknesses:**

### Strengths:
1. Clean justification and interesting idea to compress the state transition model for faster learning
2. Strong empirical results against baselines.

### Weaknesses:
1. The primary weakness I can detect is that it was not very clear to me from the text how exactly the experimental results from Tables 1 onwards are obtained. In particular, as I understand (and from looking at Section 4 Experiment Setup) first an agent is run in the environment (which agent?) in order to get samples of state transitions, then from the state transitions the empirical highway graph is constructed (according to definition 4), then the value function is learned against the simplified highway graph transitions, then that agent is evaluated. This however seems to disagree with the results in Figure 8 onwards, which have a frame budget on the x-axis, which would suggest that the highway graph is being constructed incrementally during exploration (like Algorithm 1?). I also could not find pointers in the Appendix for how exactly this is done. I had enough difficulty here that it makes it quite hard to read the rest of the text and reason about its significance, and I would very much appreciate some clarity around the matter. If it is true that there is an exploration phase before the highway graph is populated, then how were the frame budgets for Table 1 decided? If it is true that there is an exploration phase required, how important is the choice of agent for that phase?
2. I'm not the most up-to-date with model-based RL, but it would seem to me like there would be missing recent model-based baselines in the experiments. Comparing this method to model-free methods, or MuZero at the latest feels a little strange, though I would be willing to hear an argument as to why nothing more recent is suited.

Minor issues:
1. Is there are reason why Figure 15 is 3d? I don't see why simple curves wouldn't suffice.

---

> ### Author Response · Authors · 2024-08-13
> **Authors' Responses to Reviewer C93Q**
>
> Thank you for your valuable insights. We have uploaded a revised manuscript PDF with new parts highlighted in BLUE, and provide a detailed response below.
>
> Regarding the first concern:
>
> We would first briefly depict the process of agent training. 1) The training/evaluation actors are guided by the highway graph to get the experiences/episode return, whose action is taken according to the highway graph. If the actor sees an observation not in the highway graph, a random action is taken. Thus, initially, when the highway graph is empty, the actor randomly explores the environment before the first highway graph is constructed. Experimental results show the actor with random action selection during initial exploration is good enough to construct the first version of the highway graph. 2) New experiences from the agent will be used to incrementally update the highway graph, and a value learning of the highway graph will be fulfilled for better action selection. 1) and 2) are processed iteratively until the episode return of the agent is converged or reaches the maximum frames (depending on the experiment setup).
>
> From above, Table 1 records the statistics after the highway graph RL agent solved the puzzle in each maze. The experimental results from Table 1 are obtained as follows:
>
> Total Reward: the cumulative reward achieved by the agent during an episode upon convergence. The value is from the converged agent.
>
> Discounted Average Return: the average return of episodes post-convergence, adjusted for the temporal discounting factor. The value is from the converged agent.
>
> Frames Used: the total number of frames utilized by the agent prior to reaching convergence.
>
> Wall Clock Time: the total time in minutes required for the agent to converge.
>
> Converged Learning Iterations: the count of value learning iterations necessary to achieve convergence.
>
> Table 1 and Figure 8 are interrelated, with Figure 8 charting the training process as frames/time increase, while Table 1 presents the final statistics of the agent's training. For instance, in the Simple Maze 5x5 scenario, the Discounted Average Return is 0.82, mirroring the high score of the expected return of the highway graph RL agent depicted in Figure 8(b). The frame usage and wall clock time align with the figures where the performance curve transitions to convergence.
>
> The above explanation is added to the manuscript.
>
> Regarding the second concern:
>
> We agree with the reviewer's suggestion regarding the inclusion of more recent model-based baselines. Acknowledging the advancements in the MCTS-based methods after MuZero, we have elected to incorporate Stochastic MuZero as the additional baseline for comparison. Experiments are currently running and we will add them to the final manuscript once ready.
>
> For minor issues:
>
> Figure 15 has been replaced with a 2d counterpart. Please see the updated manuscript PDF.

---

> > ### Comment · Reviewer_C93Q · 2024-08-15
> >
> > Thank you for you reply. I think it has cleared up some of my confusion, though some still remains. I'll try to detail below:
> >
> > 1. How exactly are experimental results obtained
> >
> > I think the added explanation for the experiments helps a lot. Still I have questions:
> > - What exactly is meant by a "value learning iteration" here? Is it reinitializing a value function and training to convergence? Is there a neural network here or is this straight dynamic programming on a table? The common usage of value learning I see is that it proceeds asynchronously with experience collection, but this does not seem to be the case here, so I would like to understand how it is different.
> > - As I understand from your method, if the initial random exploration is unable to detect reward (for example in a tough sparse setting), then it will be difficult for the highway graph construction to help. Is this the case? It's fine if so, but I am trying to wrap my head around the method.
> > - I am still interested in knowing the frame budgets between stages for every experiment: when you decide to stop collecting experience and update the highway graph.
> >
> > 2. Additional baselines
> >
> > Thanks for adding these, I'll look forward to the additional results.

---

> ### Author Response · Authors · 2024-08-16
> **Authors' Responses to Reviewer C93Q**
>
> Thank you again for your valuable insights. We have updated the revised manuscript PDF and provided a detailed response below.
>
> > 1.1 What exactly is meant by a "value learning iteration" here? Is it reinitializing a value function and training to convergence? Is there a neural network here or is this straight dynamic programming on a table? The common usage of value learning I see is that it proceeds asynchronously with experience collection, but this does not seem to be the case here, so I would like to understand how it is different.
>
> The value learning iteration of our method refers to the iterative process of consecutively updating the values of nodes within the highway graph, by periodically utilizing newly collected experiences from the training actors. The new experiences from the training actors extend the highway graph before the updating. During the update, our method propagates the values among nodes in the highway graph based on topological adjacency, which is equivalent to dynamic programming on a table. The value function, derived from the highway graph, is considered to have **converged** only if newly collected experiences do not enhance the highway graph for better action generation. Hence, after each value learning, the highway graph will expand, and rewards from the newly collected experiences will be reflected in the actions generated from the highway graph, ultimately leading to the stabilization and convergence of the value and action from the highway graph.
>
> In the implemented framework, for the sake of simplicity, we designed the value learning and experience collection to be processed alternatively in serial not in parallel: experiences collection utilizing the highway graph version 1 &rarr; value learning to produce the highway graph version 2 &rarr; experiences collection with the updated highway graph version 2, and so forth. Moreover, it is also feasible to design a parallel asynchronous data flow for our method.
>
>
> The above explanation has been added to the updated manuscript.
>
> > 1.2 As I understand from your method, if the initial random exploration is unable to detect reward (for example in a tough sparse setting), then it will be difficult for the highway graph construction to help. Is this the case? It's fine if so, but I am trying to wrap my head around the method.
>
> Yes. The action from the highway graph guides training actors to explore.
> That means the training actors use the epsilon-greedy mechanism to probabilistically adopt the action from the highway graph, resulting in random exploration with initial directional guidance from the highway graph.
> In the case of a tough sparse setting, given that state-action values associated with a particular state are all zero, an action will be randomly selected by the highway graph.
> As a result, the training actors lose the guidance for the initial exploration direction.
>
>
> > 1.3 I am still interested in knowing the frame budgets between stages for every experiment: when you decide to stop collecting experience and update the highway graph.
>
> The training actors collect experiences by episodes for value learning, as the use of frames could potentially truncate the episode, thereby omitting crucial return information. The requisite number of episodes to be collected for each value learning across various environments is listed as follows:
> - Simple Maze 3x3: 10
> - Simple Maze 5x5: 10
> - Simple Maze 15x15: 10
> - CliffWalking: 20
> - Taxi: 20
> - Blackjack: 20
> - Custom 3 vs 2: 50
> - Custom 5 vs 5: 50
> - Custom 11 vs 11: 50
> - BattleZone: 10
> - Defender: 10
> - StarGunner: 10
>
> These details have been added to the updated manuscript.
>
> > 2. Additional baselines
>
> We have added the Stochastic MuZero and Gumbel MuZero as additional baselines to the experiment section, please see the results in the revised manuscript.

---

> ### Comment · Reviewer_C93Q · 2024-08-21
> **Thanks for your continued responses**
>
> 1.1 "What exactly is meant by a 'value learning iteration'..."
>
> Thank you for clarifying. I suspected something like this might be happening, but I wanted to make absolutely sure.
>
> 1.2 "if the initial random exploration is unable to detect reward..."
>
> Thank you for this additional clarification. It makes sense.
>
> 1.3 "I am still interested in knowing the frame budgets..."
>
> Thank you for providing these details. I believe they are useful, hence why I kept asking after them.
>
> 2. Additional baselines
>
> I see now the results of the the additional baselines and they seem even weaker than the existing ones. I'm not so familiar with these, but I looked through the descriptions in the text, and there did not seem to be much justification for why they were chosen, could you do so?
>
> I should also add that I looked at the hyperparameter tuning protocol in the Appendix, and it seems that only default hyperparameters were used for all baselines, while I assume hyperparameters were tuned for the method presented in the paper. This is very unfair to baselines. In my experience, hyperparameters for sparse-reward RL are particularly finicky (the simple maze case) and I've personally found that large batch sizes can be helpful as positive signal needs to be sampled with high probability from the replay buffer in order to propagate signal. I would appreciate some additional clarification here as to the tuning process that was undertaken for the HG results in the paper, and if it differs substantially from baselines I believe it would be necessary to spend some time tuning baselines to make results comparable.

---

> ### Author Response · Authors · 2024-09-04
> **Authors' Responses to Reviewer C93Q**
>
> We appreciate your constructive comments in helping us refine the research. We have uploaded a revised manuscript PDF with new parts highlighted in blue. In the below responses, Issue 2 will be our focal point.
>
> The rationale for incorporating Stochastic Muzero and Gumbel Muzero as new baselines is that these two algorithms have augmented Muzero in Monte Carlo Tree Search (MCTS) with numerous beneficial techniques. These include (1) Gumbel-distribution-boosted root action selection during MCTS, which significantly enhances simulation to identify superior trajectories, and (2) more adaptive transition modeling with stochastic functions to learn from the non-deterministic characteristics of environments, which has shown improved results on some real-world scenarios. We are curious about whether these enhancements may enable MCTS-based methods to understand the environment better, thereby facilitating better decisions and achieving higher performance. More specifically, we want to explore the differences and advantages by answering the following questions:
> - Does the new MCTS combined with Gumbel distribution from Gumbel Muzero better find the real value compared to our method during planning?
> - Does the stochastic transition model from Stochastic MuZero better obtain the real state of the stochastic environment?
>
> We added explanations to the new manuscript on page 12 para. 2.
>
> In response to the reviewer's comments on hyperparameter tuning, we have conducted a grid search over the most critical hyperparameters, including the experience buffer size, and size of sampling from the buffer, for Stochastic Muzero and Gumbel Muzero to ascertain the optimal settings on Maze and ToyText environments. Our results demonstrate the following:
>
> - Gumbel Muzero outperforms vanilla Muzero on Maze3x3, Maze5x5, and Cliffwalking, and is comparable on Taxi, Maze15x15, and Blackjack. This is not surprising, as Gumbel Muzero employs a novel Gumbel root action selection technique that reduces the requirement for extensive simulation to select the optimal action.
>
> - Stochastic Muzero exhibits slight improvements compared to previous results. Notably, it outperforms vanilla Muzero on Maze5x5 and is comparable on Maze15x15. Performance gaps decrease in other environments. The primary drawback is that the stochastic nature of this algorithm requires training experiences much more than 1 million frames for the model to achieve satisfactory results, as reported in their original paper.
>
> In comparison to our highway graph, the expected returns of these methods largely fluctuated in many environments and failed to obtain a high score using only 1 million frames, thus did not surpass the proposed method. MCTS within these methods present challenges in overcoming introduced noise during tree searching and the chance of obtaining biased state values from insufficient simulation. Additionally, the number of simulations to obtain the optimal value of the root state can be extremely large, but in practice, determining an adequate number of simulations is often difficult, not to mention the computational costs involved.
>
> One important factor contributing to the high performance of our highway graph is that it does not need a replay buffer, and thus no sampling is required. The highway graph itself can be considered as the buffer, and hence in each learning iteration, all information from the graph contributes to the highway graph update, since the highway graph compresses the original state-transition graph to fit into GPUs (as discussed in Section 4.2 in our manuscript). For other baselines, feeding the entire buffer into the training pipeline is intractable, due to the huge size of the buffer.
>
> Furthermore, it is worth clarifying the hyperparameters employed for the various methods described in Appendix A.
>
> - Maze and Toytext: All baselines from RLlib utilized optimal parameters that had been previously determined to be optimal through our experiments. Muzero was already fine-tuned for optimal performance. A comprehensive grid search was conducted on hyperparameters, such as buffer size, size of sampling from the buffer, number of sampling per sampling iteration, TD steps, simulations, and the network structure, for Gumbel Muzero and Stochastic Muzero, resulting in more efficient and effective code.
>
> - Football: All baselines used optimal parameters from RLlib that had been previously determined to be optimal through our experiments.
>
> - Atari: A3C, PPO, and DQN were implemented by RLlib and utilized optimal parameters from the configuration file in RLlib. NEC, GEM, EMDQN, and MFEC used default hyperparameters in the official code and also in their respective papers, which had been previously confirmed to be optimal through our experiments.
>
> We added explanations in Appendix A on page 25 para. 2.
>
> For more information regarding the modifications made to the above, please refer to the text highlighted in BLUE in the latest manuscript.

---

> > ### Comment · Reviewer_C93Q · 2024-09-10
> > **Thanks again for your continued engagement**
> >
> > Thanks for your comprehensive responses, I'll respond in detail below:
> >
> > > Rationale for Stochastic MuZero and Gumbel MuZero
> >
> > Thank you for the additional details on the baselines. I also appreciate the effort taken in tuning methods in order to make the comparisons fair. I think this greatly improves the strength of the argument. Thank you as well for the additional comment on GPU usage for the highway graph compared to replay buffer usage. I will have more on this point later.
> >
> > > Hyperparameter tuning for other baselines
> >
> > Thank you for clarifying this point. When you say "determined to be optimal through our experiments," could you describe the selection process there?
> >
> > > Extended frame budgets
> >
> > Also, given the clarification for GPU efficiency, I wanted to clarify something else: do you have a sense of the performance as the frame budget scales far beyond 1m frames? Atari frame budgets in particular can be quite large (100m frames in Rainbow [1]), at which large differences appear between methods. In Figure 11 in this work, the horizontal dashed lines are supposed to represent asymptotic performance, but the numbers in Rainbow in Table 6 (no-op starts) far surpass that with large frame budgets. Perhaps my understanding is incorrect, but It seems possible that a method like HG could be prone to early overfitting as the highway graph is constructed only from 10 episodes at a time, so if within those 10 episodes there is a better route between states that is not found, it is likely not to be found in the future as the agent navigates between existing nodes.
> >
> > I should add that my opinion of the paper is quite favorable and interesting, so I appreciate the continued dialogue.

---

> ### Author Response · Authors · 2024-09-17
> **Authors' Responses to Reviewer C93Q**
>
> We highly value your comments which have further assisted us in enhancing the research. We have uploaded the latest revised manuscript PDF, with the new sections highlighted in blue. The responses are presented as follows.
>
> >Hyperparameter tuning for other baselines
>
> For hyperparameter tuning of Rllib methods, we manually tune starting with the default hyperparameter configuration provided by the RLlib and change a selected hyperparameter with everything else fixed. The chosen hyperparameters encompass general RL hyperparameters, such as buffer size, batch size, temporal difference (TD) steps, and training iterations, alongside method-specific hyperparameters and the structure of the agent's network. To more focus on hyperparameters with critical effects during grid search, adjustments to a hyperparameter are stopped if the observed trend in expected return does not get higher.
> Through this approach, we have identified a set of hyperparameters for each method that results in the highest expected return (that we could obtain) for each respective method.
>
> We have added the explanation in Appendix A on page 25 paragraph 2.
>
> >Extended frame budgets
>
> We acknowledge your concern regarding the exploration capability within our HG RL method. Our HG RL method is capable of exploring new valuable experiences during the entire training process, and we will make it clear. It is noteworthy that while the initial highway construction is based on 10 episodes in our experiments, the training actors are designed to discover unforeseen states and transitions, thereby expanding the existing highway graph. These training actors employ varying coefficients for the epsilon-greedy mechanism, ranging from 0.1 to 0.9, indicating that there is only a 10 percent likelihood that the training actor with the lowest coefficient will be guided by the highway graph. Consequently, these training actors can accumulate more useful experiences, which can lead to a higher return compared to the current highway graph.
> This epsilon-greedy exploration strategy is extensively employed in both model-free and model-based RL methods. For instance, methods such as DQN and A3C utilize epsilon-greedy for exploration during training and rely on the trained agent network, akin to our highway graph, for decision-making during the evaluation phase.
> As a result, the exploration capability of our HG RL method is considered equivalent to that of all other baseline methods.
>
> If the frame budget is increased to 100M, the RL agents of all methods, including all baselines and our HG RL method, will have more time to explore and find trajectories with potentially higher returns for later training. In this case, the performance of all RL methods will be better than using only 1M frames.
>
> The key reasons for selecting a frame budget of 1M rather than 100M in our study are as follows:
>
> - We aim to develop a more time/frame-efficient method for propagating values within a short time and very limited frames, as opposed to many other approaches that do not consider time consumption and provide a huge frame budget. We know some RL methods were trained over billions of frames, which makes it hard to reproduce the results reported in their paper in a short time.
> - We are curious to know whether the information obtained from 1 million frames is sufficient to solve a specific task in the environment, such as Simple Maze 15x15.
> - We seek to decrease the computational expense associated with the development, training, and inference of RL methods by adopting a new approach that is suitable for deployment on smaller computing platforms, such as personal workstations, laptops, or even mobile devices, instead of using cloud-computing services from distributed clusters equipped with thousands of GPU servers, which may not always be accessible to small research groups or institutions. This motivation is in line with other recent works such as https://arxiv.org/abs/2206.11396 (TMLR 2024).
>
> We have added the explanation in Introduction on page 2 last paragraph and page 3 paragraph 2; in Experiment on page 11 last paragraph.
>
> For more details on the modifications to the above, please refer to the text highlighted in BLUE in the latest version of the manuscript.

---

> > ### Comment · Reviewer_C93Q · 2024-10-03
> >
> > Thank you for your continued clarifications, and apologies for my late response,
> >
> > > Hyperparameter tuning for other baselines
> >
> > Thank you for the clarification, I appreciate the detail and my concerns here are satisfied.
> >
> > > Extended frame budgets
> >
> > Thanks for the response. It makes sense that high epsilon would result in new early trajectory graph construction.
> >
> > I agree that the performance of all methods should be better with more experience. Still I would be curious as to the trends: does HG get more easily stuck in suboptimal trajectories due to early graph construction, or is it the case that learning is more stable and thus quicker. I understand that such training runs take a long time, and thus are not easy for fast turnaround, so it is acceptable to me not to have them for the paper, still I think it would be more informative if they were available. I certainly agree with the motivation that it is important that algorithms run much more efficiently, having worked in that space myself.
> >
> > Thank you again for your persistent engagement throughout this process, I would be happy to recommend the paper at this time.

---

> ### Author Response · Authors · 2024-10-12
> **Authors' Responses to Reviewer C93Q**
>
> We express our gratitude for all the effort and time you have dedicated to assisting us. We believe the latest comments are quite helpful and enlightening. We have further refined the contents and uploaded the latest revised manuscript PDF, with the new sections highlighted in blue. The responses are presented as follows.
>
> > Does HG get more easily stuck in suboptimal trajectories due to early graph construction, or is it the case that learning is more stable and thus quicker?
>
> To assess the exploration of our HG method, on Atari games, we reduce the number of trajectories for periodically constructing the highway graph to one-fifth of its original value and increase the frame budgets by five times (5M). Through this setting, it will be more evident whether HG is more prone to getting stuck in suboptimal trajectories. The results are presented in Figure 18 (a) - (c) (please see in the Appendix of the uploaded manuscript on page 30).
>
> According to the results, it can be observed that the expected return of HG in all three games consistently increases without getting stuck at a certain point and outperforms the performance compared to that with 1M frames.
>
> Next, we additionally assess HG on Pong. In this context, we reduce the number of trajectories for periodically constructing the highway graph to one-tenth and increase the frame budgets to 10 million. The results are presented in Figure 18 (d).
>
> From the results, HG solved Pong within 15 minutes by using approximately 6 to 9 million frames.
>
> Based on all the aforementioned results, it can be observed that HG has the ability to explore better trajectories even when the initial highway graph is extremely small.
>
> The new contents are added into the Appendix of More experimental results on page 29.
>
> For more details on the modifications to the above, please refer to the text highlighted in BLUE in the latest version of the manuscript.

---

### Review · Reviewer_aY2z · 2024-10-16

**Summary Of Contributions:**

The paper proposes a highway graph method to improve the sample efficiency of RL methods in deterministic environments with discrete state and action spaces. The paper defines the highway as a non-branching sequence of transitions can directly bring the agent from the source state to the target state. Then the paper propose to directly update the value function of the source state with the value of the target state. The paper theoretically proves the convergence to the optimality of the value learning in the highway MDP. Experiments validates the effectiveness of the propose method compared with value-based, policy-based and search-based methods on some toy and some atari games.

**Audience:**

Yes

**Claims And Evidence:**

Yes

**Requested Changes:**

I think the paper should revise according to the following points:
1.The state abstraction method also reduce the problem size and compress the state graph. However, the paper does not discuss the relationship.
2. It is hard to understand the definition the highway graph, the paper will be strengthen if the paper show some cases.
3. It is unclear how the paper parameterizes the highway graph by neural networks. There is no difference between the highway graph and the critic network.

**Strengths And Weaknesses:**

Strengths:
1.The credit assignment is a core problem of RL algorithms. The paper propose a novel method that transforms the empirical state transition  graph to the highway graph.
2. The paper proves the optimality of the method.
3. Experiments validates its effectiveness.

Weakness:
1.The state abstraction method also reduce the problem size and compress the state graph. However, the paper does not discuss the relationship.
2. It is hard to understand the definition the highway graph, the paper will be strengthen if the paper show some cases.
3. It is unclear how the paper parameterizes the highway graph by neural networks. There is no difference between the highway graph and the critic network.

---

> ### Author Response · Authors · 2024-10-26
> **Authors' Responses to Reviewer aY2z**
>
> We thank the reviewer for their comments and suggestions. We have uploaded a revised manuscript PDF with new parts highlighted in GREEN. Our responses to each comment are as follows.
>
> > The state abstraction method also reduce the problem size and compress the state graph. However, the paper does not discuss the relationship.
>
> Our highway graph is different from macro actions and other state abstraction methods in RL, since our design deliberately allows the highway RL agent to interact directly with the real environment, rather than being confined to the highway. When the environment deviates from what is recorded in the highway graph, the agent can 'fall off' the highway, gaining new experiences and incrementally expanding the highway. Consequently, this approach does not reduce the problem's size or complexity, in contrast to other state abstraction methods. The increased learning speed primarily comes from performing value propagation on the highway graph rather than on the raw state-transition graph. The updated manuscript elaborates on this point in Section 1.
>
> Changes:
> The new version added explanations on page 2 in the second paragraph.
>
>
> > It is hard to understand the definition the highway graph, the paper will be strengthen if the paper show some cases.
>
> We refer the reviewer to Figures 4 and 5 in the manuscript, which provide conceptual illustrations of highway graphs and their operation (as well as Figure 6 for highway graph construction). Briefly, the highway graph consists of intersections and paths between them, which originate from the state-transition graph of the MDP. We also show empirical highway graphs generated in our experiments in Figures 13 and 14 as additional illustrations. To further enhance the manuscript, additional explanations to show how a highway is formed have been appended in the caption and the content of Figure 13, and another case of the highway graph has been added in Figure 18 in the Appendix for the Atari game Pong.
>
> Changes:
> We have added explanations in Methodology on page 7 paragraph 4; in Experiments on page 19 paragraph 2, and Figure 13; in Appendix on page 29 paragraph 2, and Figure 18.
>
>
> > It is unclear how the paper parameterizes the highway graph by neural networks. There is no difference between the highway graph and the critic network.
>
> Our objective in parameterizing the highway graph using neural networks is that we intend to demonstrate the capacity of transforming the highway graph into a condensed and small neural function that can serve as a universal RL agent. Consequently, we train a state-action value neural function, namely the Q function, to respond to the observations from the environment. To achieve this, we utilize the collected state-action pairs and their corresponding values calculated in accordance with the highway graph as the dataset to train a simple neural network through supervised learning, in Section 3.3.3. Through this process, the information from the highway graph will be re-parameterized and added to the trained neural network. The state-action-based critic networks, concentrating on the relationship from a state-action pair to its value, are indeed identical to our trained neural network.
>
>
> Changes:
> The new version added explanations of how we parameterize the highway graph on page 11 in the last paragraph before Section 4.
>
>
> For more details on the changes in the new manuscript, please refer to the text highlighted in GREEN in the latest version of the manuscript.

---

### Review · Reviewer_EKdR · 2024-11-27

**Summary Of Contributions:**

The paper studies an effective graph-based algorithm designed to enhance the training efficiency of reinforcement learning in environments with discrete state and action spaces. The proposed approach simplifies the empirical state-transition graph by merging non-branching sequences of transitions into a single transition, referred to as a "highway." This simplification significantly reduces the number of iteration steps required during learning. The authors conducted extensive experiments across a broad range of test environments to demonstrate the effectiveness of the proposed algorithm compared to various benchmarks.

**Audience:**

Yes

**Broader Impact Concerns:**

I have no concerns on the ethical implications of the work.

**Claims And Evidence:**

Yes

**Requested Changes:**

To strengthen the work:
- It would be helpful to include discussions on other algorithms that aims to improve efficiency of reinforcement learning.

Critical for my recommendation for acceptance:
- It would be helpful to clarify in the paper if the construction of highway graph is included in performance evaluation.
- In 3.1.1, Remark 1. Could you confirm if "The number of states is reduced to z" or if the ratio compared with before is z? Could you also confirm if the reduction in time complexity will always be $\frac{1}{z^2}$? For instance, consider a scenario where the original state transition consists of a single long path made of non-branching transitions, and the Highway graph is reduced to just two states connected by a single edge.

**Strengths And Weaknesses:**

Strengths:

- The paper demonstrated an effective method to improve training efficiency of reinforcement learning by simplifying the graph structure for state transition.

- The paper presents the methodology in a detailed and clear manner.

- Extensive experiments using various environments were conducted to demonstrate the efficiency of the proposed algorithm based on the Highway graph.

Weakness:

- In 'Related Work' section, a broader discussion around other researches aiming at improving the efficiency of basic reinforcement learning algorithms could be helpful. For example, there are researches on Graph Highway Networks, Highway Value Iteration Networks, Hierarchical Reinforcement Learning, and general graph contraction/state abstraction techniques. While these approaches differ from the proposed approach, they could provide more background than basic reinforcement learning algorithms.

- In the 'Experiments' section, it is unclear whether the construction of the highway graph is incorporated into the evaluation and comparison with benchmark algorithms or if the highway graph is pre-constructed and not counted in the learning iterations.

- Since the highway consists of a non-branching sequence of transitions, the original state-transition graph is not fully connected, resulting in original complexity lower than 𝑆×𝐴×𝑆. Clarifying how this relates to the reduction of time complexity would be beneficial.

---

> ### Author Response · Authors · 2024-12-01
> **Authors' Responses (Part I) to Reviewer EKdR**
>
> We thank the reviewer for their helpful and enlightening comments and suggestions. We have uploaded a revised manuscript PDF with new parts highlighted in YELLOW. Our responses to each comment are as follows.
>
> > In 'Related Work' section, a broader discussion around other researches aiming at improving the efficiency of basic reinforcement learning algorithms could be helpful. For example, there are researches on Graph Highway Networks, Highway Value Iteration Networks, Hierarchical Reinforcement Learning, and general graph contraction/state abstraction techniques. While these approaches differ from the proposed approach, they could provide more background than basic reinforcement learning algorithms.
>
> We have added a discussion on methods aimed at improving efficiency. Furthermore, Graph Highway Networks are not grounded within the reinforcement learning context, which is not included in the discussion. The updated manuscript elaborates on this point in the related works section.
>
> Changes:
> The new version added explanations on page 4, paragraph 4; and on page 5, paragraph 2.
>
>
>
> > In the 'Experiments' section, it is unclear whether the construction of the highway graph is incorporated into the evaluation and comparison with benchmark algorithms or if the highway graph is pre-constructed and not counted in the learning iterations.
>
> In our experiments, the highway graph RL agent is constructed from scratch, which means the highway graph construction and updating are incorporated into the evaluation and comparison with benchmark algorithms.
>
> Changes:
> We have added the explanation in Experiments on page 12, the last paragraph.

---

> ### Author Response · Authors · 2024-12-01
> **Authors' Responses (Part II) to Reviewer EKdR**
>
> > Since the highway consists of a non-branching sequence of transitions, the original state-transition graph is not fully connected, resulting in original complexity lower than S×A×S. Clarifying how this relates to the reduction of time complexity would be beneficial. More precisely, in 3.1.1, Remark 1. Could you confirm if "The number of states is reduced to z" or if the ratio compared with before is z? Could you also confirm if the reduction in time complexity will always be 1/z^2? For instance, consider a scenario where the original state transition consists of a single long path made of non-branching transitions, and the Highway graph is reduced to just two states connected by a single edge.
>
> 1). z defined in Remark 1 stands for the ratio of the reduced states to the original states. We have made corrections to our language.
>
> 2). We understand the reviewer's concern about the sparsity of the reduction in time complexity. However, this sparsity of actions/edges does not influence the computational costs in our case. The reduction in time complexity will always be 1/z^2.
>
> The value iteration (VI) process aims to pass the value information among all states, which shares the value information of all states for every state. This process takes two steps: (1) interchange value information pairwisely between any pair of states, which takes |S| x |A| number of computations, where |S|, |A| is the number of states and actions. (2) pass the value information for a state from its farthest state by pairwise exchange in (1). The VI process in our work exhibits two extreme scenarios, with other cases lying in between these two.
>
> Case 1: The entire empirical state-transition graph consists of a single long path made of non-branching transitions and the highway graph is reduced to just two states, denoted as |S_inter|, connected by a single edge, as in the given scenario by the reviewer. Note that in the non-branching sequence of transitions, |A| will be 1. In addition, the possible farthest information passing starts from the last state to the initial state in this path, and the number of computations will be |S|. Therefore, the time complexity of VI on the empirical state-transition graph will be (|S| x 1) x |S|. Although the reduced states form the highway graph, the VI process does not change, and the difference is the size of the states reduced. So, the time complexity of VI for the highway graph will be |S_inter| x 1 x |S_inter|, where |S_inter| = |S| * z. That means the VI process on the highway graph will be 1/z^2 faster compared with the empirical state-transition graph.
>
> Case 2: Despite the rarity of this occurrence in RL environments, all states in the empirical state-transition graph can densely connect all the other states via an action. In this case, step (1) of VI will pass all the value information among all states, and step (2) will not be required. Since states connect to all the other states, |A| = |S|. The states are all intersections, which means |S| = |S_inter|. So, the time complexity of VI on both the empirical state-transition graph and highway graph will be |S| x |A| = |S| x |S| = |S_inter| x |S_inter| / z^2, where z = |S_inter| / |S| = 1. The condition is still satisfied that the highway graph is 1/z^2 times faster during VI.
>
> Other cases lie in between Case 1 and Case 2. Consequently, the reduction of time complexity of adopting the highway graph is in consensus with Remark 1.
>
> Changes:
> We have updated Remark 1 on page 11, the first paragraph; We added more discussions about Remark 1 in the Appendix on page 30, the second paragraph.
>
>
> Once again, we thank the reviewer for the thorough review and insightful comments. For details of the revisions made in the new manuscript, please refer to the text highlighted in YELLOW in the latest version.

---

### Decision · Action_Editor_8KQC · 2024-12-19

**Recommendation:** Accept as is

**Comment:**

Significant discussion around the experimental settings and baseline results solidified my confidence in the overall paper. The algorithm in very clearly presented and easy to grok, so once the empirical results were thoroughly discussed it made sense to accept without further revision.

**Audience:**

Improving the training efficiency of RL training, even when considered in the context of deterministic environments and finite state and actions space, should be of broad appeal to the TMLR. Particularly because impactful work in RL often comes with initial limitations that are subsequent removed (cf AlphaGo -> AlphaZero). The experimental variety ensure this method already has sufficient generality to be useful to the community at large.

**Claims And Evidence:**

This paper introduces a novel graph-based for enhancing RL training efficiency. They show gains across a wide variety of domains. There was some initial pushback as to the nature of the baselines used, but further revisions and discussion convinced the reviewers (and myself) that the baselines are logically chosen and with hyper-parameters optimized in a fair and rigorous way.